# Multicriteria assessment framework of flood events simulated with vertically mixed runoff model in semiarid catchments in the middle Yellow River

Dayang Li, Zhongmin Liang, Yan Zhou, Binquan Li, Yupeng Fu

College of Hydrology and Water Resources, Hohai University, Nanjing 210098, China

*Correspondence to*: Binquan Li (libinquan@hhu.edu.cn)

**Abstract.** Flood forecasting in semiarid regions is always poor, and a single criterion assessment provides limited information for decision making. Here, we propose a multicriteria assessment framework called flood classification–reliability assessment (FCRA) that combines the absolute relative error, flow classification and uncertainty interval estimated by the Hydrologic Uncertainty Processor (HUP) to assess the most striking feature of an event-based flood: the peak flow. One hundred flood events in four catchments of the middle reaches of the Yellow River are modeled with hydrological models over the period of 1983–2009. The vertically mixed runoff model (VMM) is compared with one physical-based model, the MIKE SHE model (originated from the Système Hydrologique Européen program), and two conceptual models, the Xinanjiang model (XAJ) and the Shanbei model (SBM). Our results show that the VMM model has a better flood estimation performance than the other models, and the FCRA framework can provide reasonable flood classification and reliability assessment information, which may help decision makers improve their diagnostic abilities in the early flood warning process.

## 1 Introduction

Arid and semiarid regions account for approximately one-third of the global land surface and half of China's land surface. A trend towards a warmer climate has increased the global incidence of intense precipitation events. Arid and semiarid regions, i.e., areas where the annual rainfall is less than 250 and 250–500 mm/a, respectively, are particularly vulnerable to this change in climate (Khomsi et al., 2016; Yatheendradas et al., 2008). More than 50% of flood-related casualties occur in these regions worldwide (Brito and Evers, 2016).

Hydrological models play an important role in flood simulation and forecasting (Devia et al., 2015). Many studies have focused on the improvement and estimation of hydrologic models in humid catchments, although there are fewer similar studies for semiarid catchments (Jiang et al., 2015). The runoff generation mechanisms for semiarid catchments are complex and may be simultaneously dominated by infiltration excess and saturation excess mechanisms (Beven, 1983; Beven and Freer, 2001).

Modeling semiarid catchments is a difficult task due to the strong spatial variability in rainfall and complexity of landscape characteristics (vegetation, soil, etc.) (Pilgrim et al., 1988). Compared with humid catchments, the rainfall of semiarid catchments is characterized by a high intensity and short duration (Andersen, 2008). In certain areas with developing economies and small populations, the raingauge networks are generally sparse. Rainfall data are important inputs for hydrologic models, and the high temporal-spatial rainfall variability combined with sparse raingauges makes modeling runoff more difficult (Hao et al., 2018; Li and Huang, 2017; Mwakalila et al., 2001).

Satellite technology has the possibility to solve the issue of low raingauge densities, although the low spatial and temporal resolutions of the products limit their applicability to subdaily rainstorms (Dinku et al., 2007). Weather radar has high spatial resolution (1 km) and temporal resolution (15 min). However, the radar costs are too high to be used for large-scale semiarid areas (Young et al., 1999).

Literature on the subdaily modeling of rainfall runoff is limited in semiarid catchments. Due to rapid times-to-peak and scarce rainfall data, capturing rainstorm flood responses is more difficult than estimating daily, monthly or annual runoff (Andersen, 2008; McMichael et al., 2006). Flood simulation results in semiarid catchments are often poor. Michaud and Sorooshian (1994) used 24 severe rainstorms that produced the largest peak flows from 1957 to 1977 to compare three hydrologic models, i.e., the lumped Soil Conservation Service (SCS) model, the simple distributed SCS model, and the distributed kinematic runoff and erosion (KINEROS) model, in the Walnut Gulch catchment, and none of them were able to accurately simulate flood events. McIntyre and Al-Qurashi (2009) analyzed 27 flood events with three hydrologic models, the lumped Identification of Hydrographs and Components from Rainfall, Evaporation and Streamflow (IHACRES) model, the distributed IHACRES model, and a 2-parameter regression model in a catchment in Oman. The average absolute relative errors in the flow peak and flow volume were 53% and 36%, respectively, for the best performing models. Under current technical conditions, it seems difficult to achieve an acceptable simulation/forecasting result for flood events in semiarid catchments. Therefore, determining how to use modeling results with limited accuracy to provide guidance for early flood warnings is important.

In this study, a multicriteria assessment framework that combines the absolute relative error, flow classification and the uncertainty interval estimated by Hydrologic Uncertainty Processor (HUP) is proposed to provide more information for engineers' decision making. Four hydrological models, the vertically mixed runoff model (VMM), the MIKE SHE model (originated from the Système Hydrologique Européen program), the Xinanjiang model (XAJ) and the Shanbei model (SBM), are compared based on the performance of the modeling results in four catchments in the middle reaches of the Yellow River. The global sensitivity analysis (GSA) method PAWN is used to analyze the parametric sensitivity of the VMM model. The rest of the paper is organized as follows. Section 2 describes the study area and the data set used. Section 3 presents the VMM model methodology, model set, model calibration and validation, multicriteria assessment framework and parameter sensitivity analysis. Section 4 describes the results and discussion of model comparison, sensitivity analysis and analysis of the

multicriteria assessment framework for the VMM model. The final section presents the conclusions of the study.

**2 Study area and data**

The four selected study catchments are all key tributaries located in the middle reaches of the Yellow River, China (Fig. 1). The maximum and minimum areas of the catchments are 1989 km$^2$ and 8706 km$^2$, respectively. The average annual temperature ranges from 6 to 14 °C. The average annual precipitation ranges from 1010 to 1150 mm, and 65 to 80% is concentrated in summer (Li et al., 2019; Li and Huang, 2017; Xiao et al., 2019). The rainfall is generally characterized by high intensity and short duration. The average annual evaporation ranges from 1010 to 1150 mm. All selected catchments are semiarid based on an aridity index between 2.31 and 2.78 (UNEP, 1992). This catchment information is listed in Table 1.

The lack of vegetation in these catchments leads to severe soil erosion, and the average sediment concentration reaches 126 kg m$^{-3}$ according to Li et al. (2019). Some hydrologists have studied daily and monthly rainfall runoff, although few studies have modeled hourly floods. With the rapid increase in population and economic development, flood disasters have received increasing attention. Hence, it is important for decision makers to know how to evaluate the flood risk when a flood is approaching.

The period used in the model was from 1983 to 2009. Continuous streamflow and rainfall data were collected from streamflow gauging stations and rain gauging stations at a daily time step, respectively; streamflow and rainfall data for each of the flood events were collected at an hourly time step. Nine rainfall gauging stations in the Qiushui River catchment, 15 rainfall gauging stations in the Qingjian River catchment, 12 rainfall gauging stations in the Tuwei River and 41 rainfall gauging stations in the Kuye River were selected. The Thiessen polygon method was used to interpolate the rainfall data for each catchment.

**3 Methodology**

**3.1 Vertically mixed runoff model**

The VMM is a lumped, continuous hydrologic model and has been used in many areas in China, especially in semiarid and subhumid catchments (Bao and Zhao, 2014; Li, 2018; Wang and Ren, 2009). Compared with other conceptual models, such as the XAJ model (Zhao, 1992) and the Sacramento Soil Moisture Accounting (SSMA) model (Burnash et al. 1973), among others, the VMM model is capable of simulating the saturation excess and infiltration excess runoff generation mechanisms simultaneously. As shown in Fig. 2, the VMM model combines the infiltration capacity curve and tension water content storage capacity curve in the vertical direction. Net rainfall (observed rainfall after removal of evaporation, *PE*) is partitioned into surface runoff (*RS*) and infiltration flow (*FA*) by the infiltration capacity curve in the VMM model. *FA* is regulated by the tension water storage capacity curve, part of which supplements the tension water storage (*W*), with the remainder of the

rainfall forming groundwater flow (*RB*) (including unsaturated flow and saturated flow). Here, the calculation of runoff generation is described briefly. More detailed information about the VMM model is contained in Bao and Zhao (2014).

The improved Green-Ampt infiltration curve (Bao, 1993) is applied in the VMM model as the infiltration capacity curve, and the equation is as follows:

$$FM = FC\left(1 + K\frac{WM - W}{WM}\right) \tag{1}$$

where *FM* is the average point infiltration capacity of the catchment, and the descriptions of *WM*, *K*, and *FC* are shown in Table 2.

*FA* is calculated by Eq. (2):

$$FA = \begin{cases} FM - FM\left(1 - \frac{PE}{(FMM)^{1+BF}}\right) & PE < FMM \\ FM & PE \geq FMM \end{cases} \tag{2}$$

where

$$FMM = FM(1 + BF) \tag{3}$$

in which *FMM* is the maximum point infiltration capacity of the catchment and *BF* is defined in Table 2.

The part that exceeds the average point infiltration capacity of the catchment *FM* forms *RS*. *RS* can be calculated by Eq. (4).

$$RS = PE - FA \tag{4}$$

*RB* can be calculated by Eq. (5):

$$RB = \begin{cases} FA - WM + W + WM\left(1 - \frac{W^* + FA^{B+1}}{WMM}\right) & FA + W^* < WMM \\ FM - WM + W & FA + W^* \geq WMM \end{cases} \tag{5}$$

where

$$W^* = WMM\left[1 - \left(1 - \frac{W}{WM}\right)\right]^{\left(\frac{1}{B}+1\right)} \tag{6}$$

$$WMM = WM(1 + B) \tag{7}$$

in which *WMM* is the maximum point tension water storage capacity of the catchment; $W^*$ is the ordinate of Fig. 2b; which represents the point tension water content capacity in the catchment; and *B* is defined in Table 2.

The outlet runoff *R* can be calculated as follows:

$$R = RS + RB \tag{8}$$

**3.2 Model set of the VMM model**

The VMM model was run continuously from 1983 to 2009 for each catchment. Rainfall data were available only at an hourly time step over the periods of flood events, and for other periods, they were available at a daily time step. Hence, the time step

of simulations was set as daily between flood events and hourly within flood events for each catchment. To consider the spatial variation in rainfall, the subcatchments were divided according to the stream networks, and each subcatchment contained at least one rainfall gauging station. The areal mean rainfall of each subcatchment was calculated using the Thiessen polygon method. Because streamflow data were only available in the outlet streamflow gauging station for each catchment, the spatial variation in each catchment's parameters could not be determined by calibration. Thus, the parameters (Table 2) were set uniformly in all subcatchments. Two initial values, the initial tension water storage (*W0*) and the initial free water storage (*S0*), were used to describe the initial catchment moisture condition. The initial values are smaller for drier catchments, and the minimum values are zero. In this study, the initial values were assumed to be zero uniformly due to the dry conditions at 00:00:00 on January 1, 1983 for each catchment. It should be noted that continuous simulations for each catchment eliminate the need to set the initial values for each flood event in a catchment.

## 3.3 Model calibration

The fourteen parameters (Table 2) of the VMM model were calibrated using the Shuffled Complex Evolution (SCE-UA) global optimization algorithm (Duan et al., 1993). The ranges of parameters were determined based on previous literature and prior knowledge (Bao and Zhao, 2014; Li et al., 2018). Due to the rapid rise and fall of floods (usually less than 24 h) in semiarid catchments, accurate simulations of the full hydrograph are not needed and cannot be achieved. The Nash-Sutcliffe efficiency (NSE) (Nash and Sutcliffe, 1970) is widely used as an objective function of calibration in humid catchments; however, it may not be suitable for semiarid catchments because a good fit is not required between the simulated and observed streamflows. McIntyre and Al-Qurashi (2009) and Sharma and Murphy (1998) used the absolute relative error to evaluate model outputs (flow peak and flow volume) for semiarid areas, and the calibrated results indicated that the peak flow results are more accurate than the suggested results based on the NSE. Thus, the simulated hydrograph is reasonable for the majority of flood events. The equations are as follows:

$$E_p = \frac{1}{n} \sum_{i=1}^{n} \frac{\left| Q_p^i - Q_{pm}^i \right|}{Q_{pm}^i} \tag{9}$$

$$E_v = \frac{1}{n} \sum_{i=1}^{n} \frac{\left| Q_v^i - Q_{vm}^i \right|}{Q_{vm}^i} \tag{10}$$

where $E_p$ and $E_v$ are the average performances (in terms of absolute relative error) for peak flows and flow volumes in each catchment, respectively; $n$ is the number of events; the index $i$ denotes each event; $Q_p$ and $Q_{pm}$ are the simulated and measured values of peak flow per event, respectively; and $Q_v$ and $Q_{vm}$ are the simulated and measured values of flow volume per event,

respectively.

Constraining the model output with peak flows and flow volumes can be expressed as follows:

$$E_{pv} = \frac{E_p + E_v}{2} \tag{11}$$

where $E_{pv}$ is the objective value. The model outputs become better as the value of $E_{pv}$ approaches 0. The number of iterations was set to 2000 in the calibration process.

**3.4 Model comparison**

To achieve a better performance in rainstorm flood simulations, three hydrologic models, including two conceptual models, XAJ and SBM, and one distributed model, MIKE SHE, were used for comparison with the VMM model. The XAJ model was developed by Zhao (1992) and has a single saturation excess runoff generation mechanism. The XAJ model has been successfully applied in humid and subhumid catchments (Cheng et al., 2006; Lü et al., 2013). The SBM model was developed by Zhao (1983) and has a single infiltration excess runoff generation mechanism. The SBM model is generally used in semiarid or arid catchments in China (Bao et al., 2017; Li and Zhang, 2008; Zhao et al., 2013). In addition, the MIKE SHE model is a deterministic, physically-based distributed hydrologic model that can simulate surface water flow, unsaturated flow and saturated flow (Jayatilaka et al., 1998). The MIKE SHE model has been used to solve water resources and environment problems at different spatiotemporal scales (Li et al., 2018; Rujner et al., 2018; Samaras et al., 2016).

**3.5 Multicriteria assessment framework: flood classification–reliability assessment for flood events**

Flood simulations and forecasting in semiarid catchments are very difficult due to strong spatially the variability of rainfall, complex landscape characteristics and others. Although some hydrologists improve flood simulations and forecasting by improving hydrologic models, the improvements are always limited or are suitable for only specific regions (Collier, 2007). The flood peak is the most significant feature in semiarid regions. Determining the extent to which the calculation of flood peaks can be accepted is crucial. Generally, the absolute relative error is used to measure the calculation of flood peak accuracy; for example, 20%, 30% or similar values are acceptable (Li et al., 2014; McIntyre and Al-Qurashi, 2009). To provide more information for flood defense management, the generalized likelihood uncertainty estimation (GLUE) and the Bayesian framework with Markov Chain Monte Carlo sampling are used to provide probabilistic forecasting, such as the 95% uncertainty interval (Christiaens and Feyen, 2002; Li et al., 2017), although these methods may not lead to clear decisions (Beven, 2007).

In this study, to obtain a better diagnostic and discriminatory method for the decision maker, we propose a multicriteria assessment framework called the flood classification—reliability assessment (FCRA) in the catchments of the middle reaches of the Yellow River. The FCRA framework consists two parts: i) flood classification and ii) flood reliability assessment. The first part represents floods are classified with percentiles and the absolute relative error; the other represents the reliability of

flood modeling is evaluated with the Bayesian method. Peak flows, as the most prominent features of flood events, are assessed with the FCRA framework. Detailed descriptions can be found as follows:

(C1) The absolute relative error of peak flow should be less than 20%.

(C2) The modeled and observed peak flows should be in the same flow zone: the observed peak flow $Q_p$ for all flood events in a catchment are divided into three zones (low flow zone, medium flow zone, high flow zone), with 25th percentiles $Q_{p25}$ and 75th percentiles $Q_{p75}$ as the boundary points; if $Q_p \leq Q_{p25}$, then the peak flow $Q_p$ belongs to the low flow zone; if $Q_p \geq Q_{p75}$, then the peak flow $Q_p$ belongs to the high flow zone; the remaining flow peaks belong to the medium flow zone. Both the 25th percentile and 75th percentile are commonly used to distinguish zones.

(C3) The observed peak flows should fall within one standard deviation (σ) of the mean (approximately 68.3% uncertainty interval) peak flow estimated by the Hydrologic Uncertainty Processor (HUP), one component of the Bayesian forecasting system detailed in Krzysztofowicz (1999) and Biondi et al. (2010).

Conditions C1 and C2 are flood classification criteria. If the observed and modeled peak flows meet one of the two conditions, it is believed that they are the same types of floods. The key of the FCRA framework is condition C2, and condition C1 is used to avoid errors caused by flow zone boundaries. For example, when $Q_{p75} = 200$ m³/s, the modeled peak flow equals 198 m³/s and the observed peak flow equals 201 m³/s. However, using only condition C2 may lead to inappropriate model results; adding condition C1 can help address the problem. Condition C3 is used to assess the reliability of peak flows modeling; small uncertainty interval (68.3%) is used that has narrow upper and lower limits. This interval may reduce the numbers of observed peak flows that fall within the confidence level. A modeled peak flow that can be accepted should satisfy condition C1 or condition C2 and then condition C3.

**3.6 Parameter sensitivity analysis**

A sensitivity analysis (SA) (Saltelli et al., 1989) was proposed to assess the effects of inputs on the model output. The SA can be classified into a GSA and local sensitivity analysis (LSA). Compared with the LSA, the GSA is capable of analyzing the effects of inputs within the entire input domain. The Fourier amplitude sensitivity test (Cukier et al., 1973), Sobol method (Sobol, 1993) and Morris screening method (Morris, 1991) are the most widely used GSA methods in the assessment of parameter sensitivity in hydrologic models. Pianosi and Wagener (2015) proposed the novel GSA method PAWN (derived from the authors' names), which is based on the cumulative density function. PAWN has advantages over the parameter ranking and time-consuming nature of other GSA methods (Khorashadi et al., 2017). In this study, we used the PAWN method to perform a GSA on the VMM model.

Considering $x_{i,j}$ ($i, j = 1, 2, \cdots$, where $i$ and $j$ represent the $i$-th input parameters and the $j$-th sampling, respectively) as sensitivity inputs, then the sensitivity of $x_{i,j}$ can be measured by the distance between $F_{(y_i|x_{i,j})}(y_i)$ (the cumulative probability distribution function of $y_i$ when $x_{i,j}$ changes between the upper bound and lower bound) and $F_{y_i}(y_i)$ (the

cumulative probability distribution function of $y_i$ when $x_i = \frac{1}{n}\sum_{j=1}^{n} x_{i,j}$, where $n$ is the number of samplings per input parameter). The Kolmogorov–Smirnov statistic (Simard and Ecuyer, 2011) is used to measure the distance between $F_{(y_i|x_i)}(y_i)$ and $F_{y_i}(y_i)$:

$$KS(x_{i,j}) = \max_{1 \le j \le n} \left| F_{y_i}(y_i) - F_{(y_i|x_{i,j})}(y_i) \right| \tag{12}$$

As $KS$ varies with $x_{i,j}$, the maximum of all possible $KS$ values is included in the PAWN index $P_i$:

$$P_i = \max_{1 \le j \le n} KS(x_{i,j}) \tag{13}$$

$P_i$ ranges from 0 to 1 and $x_i$ becomes more sensitive as $P_i$ approaches 1. A $P_i$ equal to 1 indicates that $x_i$ has no effect on the model. For more information about PAWN, please refer to Pianosi and Wagener (2015). In this study, the number of evaluations was set to 500, as suggested by Pianosi and Wagener (2018).

**3.7 Model validation**

The modeling period was between 1983 and 2009. In the Qiushui River, 20 flood events were selected, with the first 15 events used for calibration and the remaining five events used for validation. Similarly, in the Qingjian River, 29 flood events were selected, with 24 events used for calibration and the remaining five events used for validation. In the Tuwei River, 23 flood events were selected, with 18 events used for calibration and the remaining five events used for validation. Finally, in the Kuye River, 28 flood events were selected, with 23 events used for calibration and the remaining five events used for validation.

**4 Results and discussion**

**4.1 Comparison of model results**

Boxplots of the absolute relative errors of the peak flows for each model in the four catchments are shown in Fig. 3. In terms of the median and average of the absolute relative errors for peak flows, the VMM model has the lowest values for both calibration and validation in Figs. 3a–g except for the validation period in the Kuye River catchment in Fig. 3h; in most cases, the MIKE SHE model has lower median and average values than the XAJ and SBM models, i.e., Figs. 3a, b, c, d, g, h. Low median and average values indicate that more modeled flood events have good performance in a catchment. Except for the good performance in the Tuwei River catchment, the results using the SBM model are as poor as those using the XAJ model in the other catchments. In terms of interquartile ranges (IQR) of the absolute relative errors for peak flows, the VMM and MIKE SHE models have relatively small ranges (Figs. 3a, c, d, g) and the SBM and XAJ models have large ranges in most cases (Figs. 3a, b, c, d, g). This indicates that the VMM and MIKE SHE models are more robust to reproduce the peak flows in the middle reaches of the Yellow River.

Tables 3 and 4 show the average performance in terms of the absolute relative error for flow volume $E_v$ and the lag time for the four models in each catchment, respectively. Low $E_v$ and lag time values indicate that the model is highly capable of

reproducing the flow volumes and time-to-peak values. The VMM model has the minimum average $E_v$ and lag time, with values of 39.01% and 3.05 h, respectively (Tables 3 and 4). In contrast, the XAJ model has the maximum average $E_v$ and lag time, with values of 58.93% and 4.51 h, respectively. The MIKE SHE and SBM models have similar performances in terms of average $E_v$ and lag time.

The analysis of Fig. 3, Table 3 and Table 4 above indicates that the VMM model has the best performance to reproduce the peak flows, flow volumes and lag times in the four studied catchments of the middle reaches of the Yellow River and the XAJ model has the worst performance. In addition, the MIKE SHE model is superior for reproducing the peak flows but exhibits similar performance compared with the SBM model for reproducing the flow volume and lag time. Although the MIKE SHE model is a distributed hydrologic model with more complex structures and more explicit physical meaning than the conceptual VMM model, it does not achieve better results than the conceptual VMM model due to a lack of sufficiently high-resolution data, and this is consistent with other studies (Beven, 2002, 2011; Michaud and Sorooshian, 1994; Seyfried and Wilcox, 1995). Both infiltration excess and saturation excess can be simulated via the VMM model; it may be the reason why it performs better than the other two conceptual models (XAJ and SBM), which have single runoff generation mechanisms (saturation excess and infiltration excess, respectively).

**4.2 Sensitivity analysis of the VMM model**

The GSA method PAWN is applied to estimate the influence of parameter uncertainty on the model output results. Figure 4a and Figure 4b show the average SA results of all study catchments for the objective function $E_p$ (Eq. 9) and $E_{pv}$ (Eq. 11), respectively. The parameters become more sensitive as the ranking becomes higher. Parameters *CS*, *IM* and *KE* have the highest rankings whether the objective function of the VMM model is $E_p$ or $E_{pv}$. The rankings of other parameters are influenced slightly by different objective functions, such as *CG*, except for *WM*. *WM* ranks sixth when $E_{pv}$ is the objective function and 12th when $E_p$ is the objective function. This ranking is because *WM* controls the tension water content in the soil, which determines the amount of rainfall stored in the soil and the generation of runoff. There may be a strong relationship between flow volume and *WM*. Therefore, *WM* has a higher ranking when the objective function considers the effect of flow volume.

**4.3 Flood classification—reliability assessment of the VMM model**

The FCRA framework we propose is applied to assess the ability of the VMM model to model flood events in the four catchments. FCRA requires that the accepted modeled peak flows have the same flood types (high flow, medium flow or low flow) as the observed peak flows; in addition, the modeled peak flows should be reliable. Similar types of peak flows that represent the modeled peak flows should meet one of the requirements of conditions C1 and C2; the modeled peak flows that are reliably represented need to meet condition C3. The observed peak flows and the modeled peak flows under condition C1, C2 or C3 are shown in Figure 5. The percentages of modeled peak flows that meet the conditions are presented in Table 5.

Although the percentages of the modeled peak flows that meet condition C1 are less than 50% (Table 5), they reduce the boundary effects of flood classification. Taking the 13th flood event of the Kuye River catchment as an example, the observed and modeled peak flows are 1230 m$^3$/s and 1510 m$^3$/s, respectively. As shown in Fig. 5d, the absolute relative error for peak flow is greater than 20%. In addition, for the Kuye River catchment, it is reasonable to believe that the peak flows 1230 m$^3$/s and 1510 m$^3$/s may have the same risk according to the known flood peak data, which can be classified as the same flood type (medium flow) according the condition C2.

From the Table 5, we find that 95% or more of modeled peak flows meet condition C3; this indicates that almost all modeled peak flows have less uncertainty and more reliability in the selected catchments. Figure 5 shows more directly that the majority of peak flows for the observations and modeling fall between the 15.85th percentile and the 81.45th percentile (68.3% uncertainty interval) estimated by HUP, which is consistent with Table 5. In addition, the percentages of modeled flood events and observed peak flows that are the same flood types (shown in Table 5 with C$^*$) equal the acceptance rate (shown in Table 5 with C$^{**}$) for each catchment due to the high reliability of modeled peak flows. Under the FCRA framework, the acceptance rates (C$^{**}$) for the catchments are more than 65% except for the Qingjian River catchment. This indicates that the FCRA framework may have the diagnostic capability to assess the modeled flood events in the four semiarid catchments.

Under the FCRA framework, a modeled flood event could be assessed to determine what flood type (high flow, medium flow, low flow) it is and how reliable it is. This information is meaningful in the early flood warning process in the semiarid catchments. Although FCRA is simple and even coarse, it is convenient and beneficial in helping engineers make decisions when a flood is approaching.

**5 Conclusions**

In this study, a multicriteria assessment framework of flood events called the flood classification—reliability assessment (FCRA) is proposed with the VMM model in four semiarid catchments of the middle reaches of the Yellow River. The main conclusions are as follows.

(1) Compared with the distributed model MIKE SHE and the two conceptual models, XAJ and SBM, the VMM model has a better performance for modeling flood events in the middle reaches of the Yellow River.

(2) In the four catchments, the parameters confluence coefficient of surface flow (*CS*), impermeable area (*IM*), and residence time of Muskingum (*KE*) in VMM model are the most sensitive parameters based on an analysis by the global sensitivity method PAWN; in addition, the sensitivity ranking of the parameter *WM* related with the soil moisture capacity is the most affected by the objective functions.

(3) The FCRA framework combining flood classification and reliability assessment may have the reliable diagnostic capability to assess flood events in the early flood warning process. It should be noted that condition C2, which divides

peak flows into three flow zones, will be affected by the number of observed peak flows when data availability is limited. The framework is suitable for semiarid regions with poor modeling results and provides guidance for decision making.

**Code availability**

We have shared the MATLAB code for the VMM model at https://doi.org/10.4211/hs.c5232287d5c04bfb8cac5ce4e391ea0f.

**Author contributions**

DL wrote the text and developed the MATLAB code for the VMM model. DL, ZL, YZ and BL conceived the study. All co-authors jointly worked on improving the text and responded to the editor's and the reviewers' suggestions.

**Competing interests**

The authors declare that they have no conflict of interest.

**Acknowledgments**

This study was supported by the National Key Research and Development Program of China (grant no. 2016YFC0402706), the National Natural Science Foundation of China (grant nos. 41730750, 41877147), and Special Scientific Research Fund of Public Welfare Industry of Ministry of Water Resources, China (201501004), sponsored by Qing Lan Project. We would like to thank Francesca Pianosi (University of Bristol) for providing the program code of PAWN at https://www.safetoolbox.info

15 /pawn-method/. We also thank the editor and the anonymous reviewers, whose comments have largely improved this work.

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

**Table 1. Characteristics of the four catchments**

| Catchment | Area (km$^2$) | Outlet station | Area[*] (km$^2$) | Mean annual precipitation (mm) | Mean evaporation (mm) | Aridity index |
|---|---|---|---|---|---|---|
| Qiushui River | 1989 | Linjiaping | 1873 | 499 | 1150 | 2.31 |
| Qingjian River | 4080 | Yanchuan | 3468 | 451 | 1080 | 2.4 |
| Tuwei River | 3294 | Gaojiachuan | 2095 | 377 | 1050 | 2.78 |
| Kuye River | 8706 | Wenjiachuan | 8645 | 410 | 1010 | 2.46 |

[*] The area of the catchment controlled by the outlet station indicated in the table.

**Table 2. Parameter values of the VMM**

| Symbol | Definition | Range* |
|--------|------------|--------|
| KC | Ratio of potential evapotranspiration to pan evaporation | [0.5, 1.5] |
| WM | Mean areal maximum possible soil moisture, mm | [50, 200] |
| FC | Stable infiltration capacity, mm/h | [5, 100] |
| K | Infiltration index related to soil permeability, /h | [0.05, 1] |
| BF | Index of the watershed infiltration capacity curve | [0, 0.5] |
| B | Index of the watershed water storage capacity curve | [1, 2] |
| KI | Outflow coefficient of interflow, d | [0.1, 0.5] |
| KG | Outflow coefficient of groundwater, d | [0.5, 2] |
| CS | Confluence coefficient of surface flow | [0.05, 0.9] |
| CI | Recession coefficient of interflow, d | [0.5, 0.95] |
| CG | Recession coefficient of groundflow, d | [0.90, 0.99] |
| KE | Residence time of Muskingum, h | [0.5, 5] |
| XE | Muskingum coefficient | [0.01, 0.49] |
| IM | Impermeable area | [0, 1] |

*In [a, b], a and b represent the lower and upper bounds of the parameters, respectively.

**Table 3. Performance (in terms of absolute relative error) for peak flow $E_v$ in each catchment in the four models**

Unit: %

| | Qiushui River | Qingjian River | Tuwei River | Kuye River | Average[*] |
|---|---|---|---|---|---|
| VMM | 26.52 | 58.50 | 40.20 | 30.80 | 39.01 |
| MIKE SHE | 40.50 | 60.70 | 45.30 | 38.20 | 46.18 |
| XAJ | 56.60 | 66.61 | 60.20 | 52.30 | 58.93 |
| SBM | 38.14 | 55.82 | 35.50 | 45.2 | 43.15 |

[*]The average $E_v$ of the four catchments for each model

**Table 4. Lag time of the peak flow in the four catchments in the four models**   Unit: h

| | Qiushui River | Qingjian River | Tuwei River | Kuye River | Average[*] |
|---|---|---|---|---|---|
| VMM | 2.20 | 3.02 | 3.46 | 3.50 | 3.05 |
| MIKE SHE | 2.50 | 3.50 | 4.20 | 3.90 | 3.53 |
| XAJ | 4.10 | 3.81 | 5.62 | 4.50 | 4.51 |
| SBM | 4.00 | 2.95 | 3.46 | 4.20 | 3.65 |

[*]The average lag time in the four catchments for each model

**Table 5. The percentage of modeled peak flows that meets various conditions for the VMM model**

Unit: %

| | C1 | C2 | C3 | C* | C** |
|---|---|---|---|---|---|
| Qiushui River | 40.00 | 75.00 | 95.00 | 75.00 | 75.00 |
| Qingjian River | 41.38 | 44.83 | 100.00 | 58.62 | 58.62 |
| Tuwei River | 47.83 | 69.57 | 100.00 | 69.57 | 69.57 |
| Kuye River | 35.71 | 64.29 | 100.00 | 67.86 | 67.86 |

C* represent the modeled peak flows that meet one of the conditions C1 and C2; this means modeled and observed peak flows are the same type

C** represent the modeled peak flows that meet the conditions C* and C3

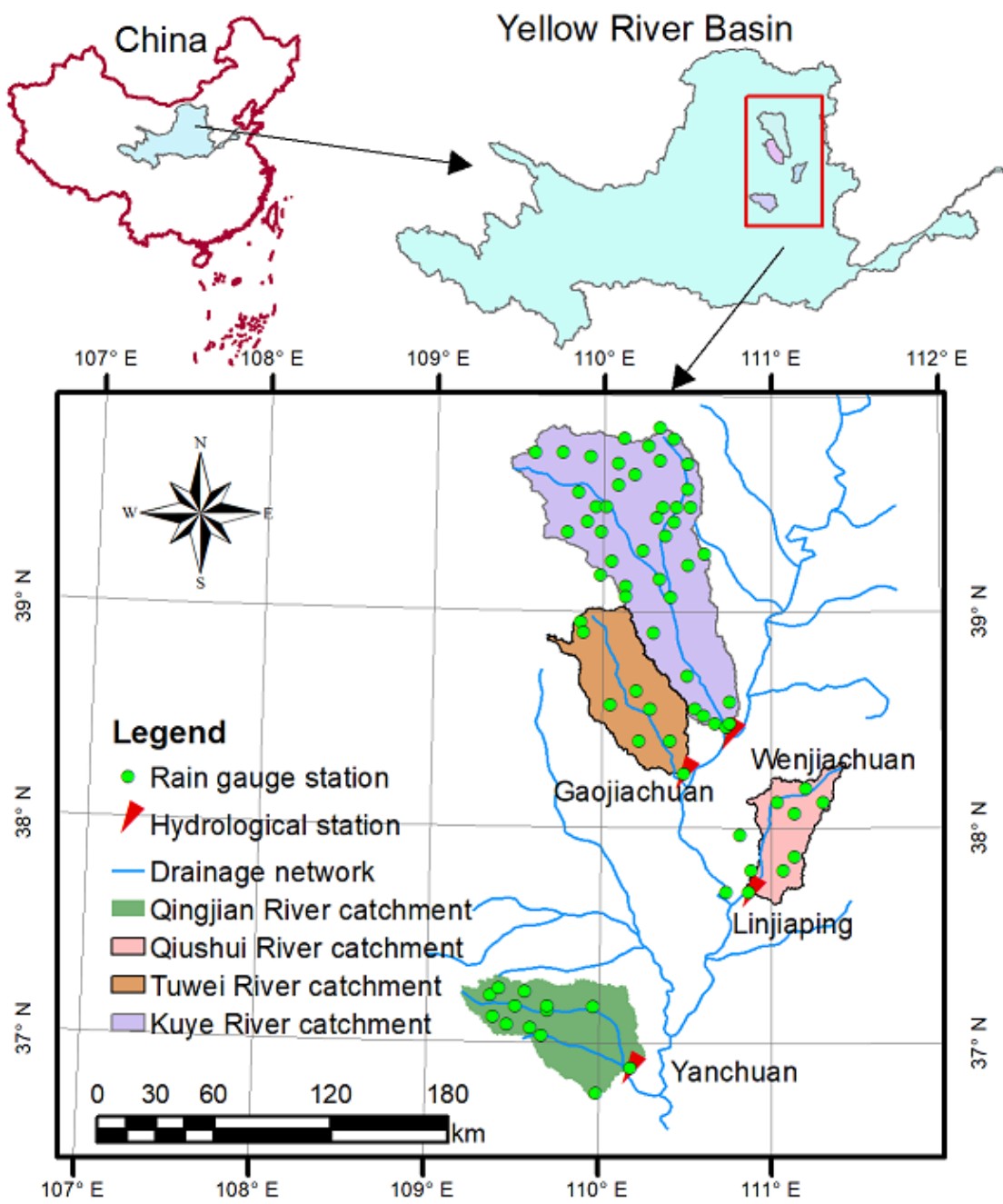

**Figure 1: Locations and raingauging nets of the Qingjian River catchment, Qiushui River catchment, Tuwei River catchment and Kuye River catchment.**

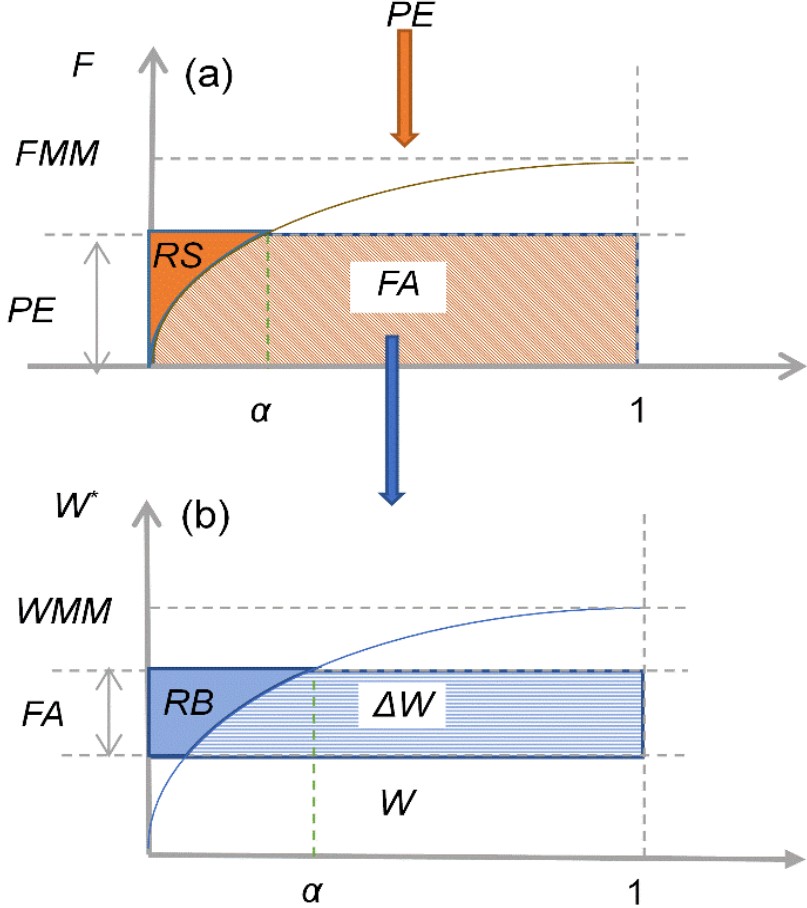

**Figure 2: Runoff generation module in the VMM model. (a) Infiltration capacity curve; and (b) tension water content storage capacity curve. $\alpha$ is the fracture area that is saturated and $F$ represents the point infiltration capacity.**

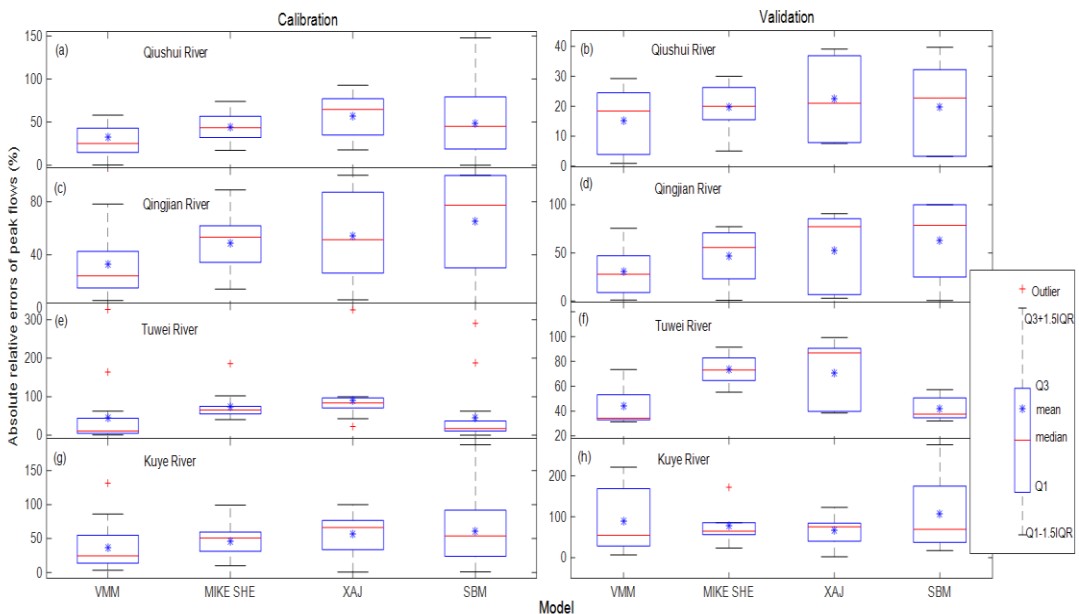

**Figure 3: Boxplot of absolute relative errors of peak flows in the four catchments. Q1 and Q3 represent the first quantile and third quantile, respectively; interquartile range (IQR) = Q3 − Q1. An outlier is defined as an extreme value that exceeds the range of Q1 − 1.5 IQR and Q3 + 1.5IQR.**

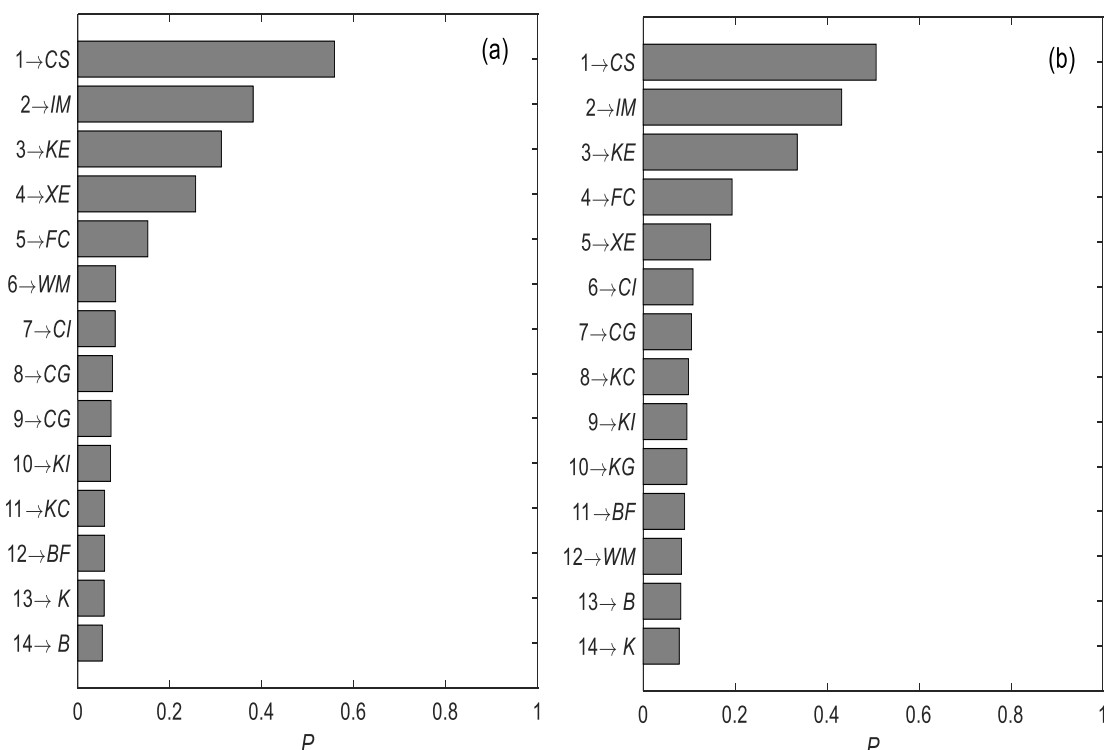

**Figure 4: Sensitivity rankings of the VMM parameters based on the global sensitivity analysis method PAWN for different objective functions. (a) $E_{pv}$ as the objective function and (b) $E_p$ as the objective function. The value $P$ is used to assess the sensitivity degree of the parameter with the PAWN method, and a larger value corresponds to greater sensitivity. The numbers on the ordinate represent the sensitivity rankings.**

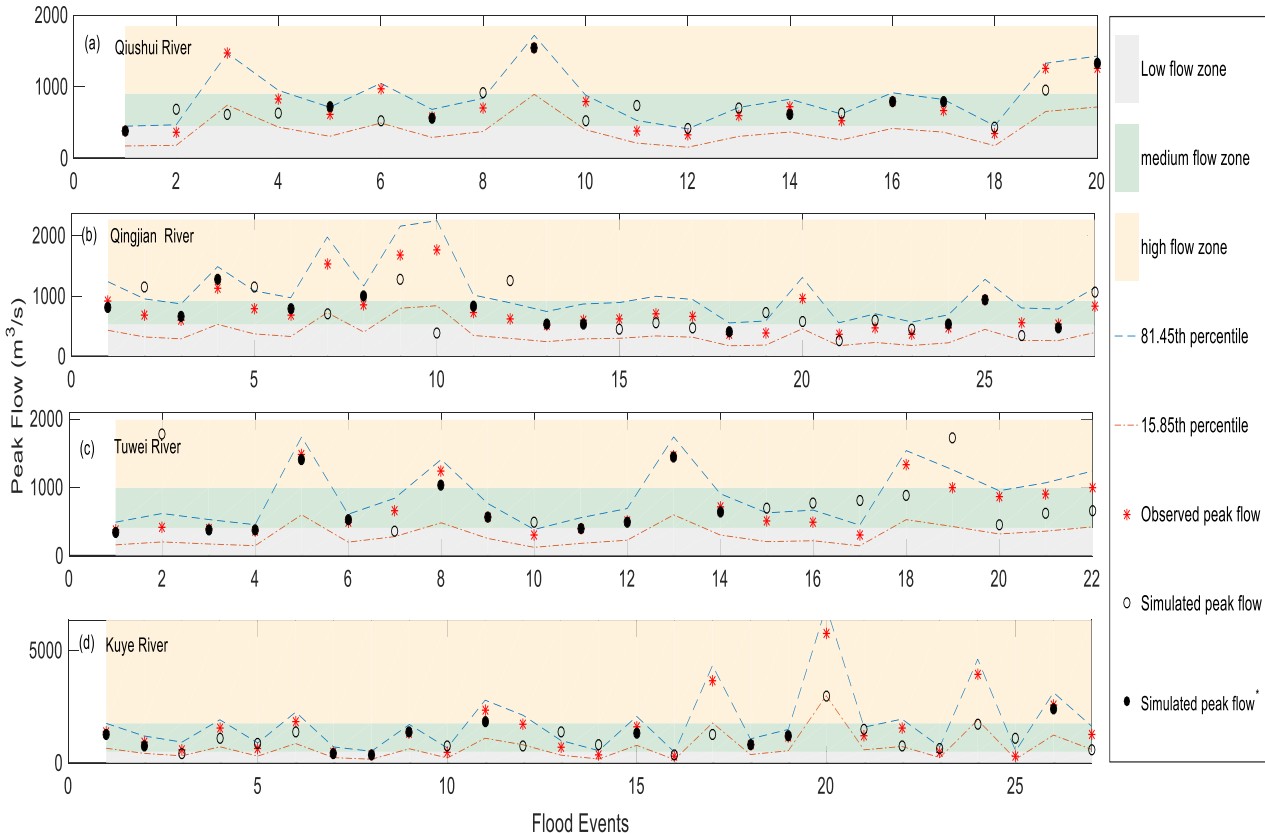

**Figure 5: Observed peak flows (red asterisk) and simulated peak flows (circles) with the VMM model for each catchment under conditions C1, C2 and C3. Flood peaks conforming to condition C1 are represented by solid circles and the others are empty; the three flow zones (low, medium and high flow zones) classified by condition C2 are shown in gray, green and off-white, respectively; the 68.3% uncertainty interval of peak flows estimated by condition C3 is shown between the blue dashed line (81.45th percentile) and the red dashed-dotted line (15.85th percentile).**