# Peer review of "Multicriteria assessment framework of flood events simulated with vertically mixed runoff model in semiarid catchments in the middle Yellow River"

_Natural Hazards and Earth System Sciences, 2018_

## Referee Comment (RC1) · Anonymous Referee #1 · 11 Mar 2019

The paper presents a multicriterion assessment framework for flood events forecasting or warning in semi-arid regions. Four hydrologic models have been used in catchments of the middle Yellow River. The result shows that the VMM model has a better performance of flood modeling than the other three models. The authors believe that flood events in semi-arid and arid regions should have different criteria than that of in humid areas to determine whether a flood forecasting and early warning is acceptable. The topic of this study is very interesting and the idea is more or less novel. The paper is well-structured. I recommend the publication of this paper with a minor revision. Some

specific comments are listed as below. Specific comments: (1) The proposed framework has three parts, C1, C2 and C3. C2 is the key part of this framework, and three flow zone (low flow zone, medium flow zone, high flow zone) are divided. I think this simple framework is more important in terms of flood early warning rather than just a performance assessment. This may be real value of the framework but it is not clearly expressed in the manuscript. More explanation and discussion should be added in the paper. (2) The initial condition is very important for a hydrologic model. In this paper, it is reasonable that the daily based model is used to calculate the initial conditions of the event-based model, but the initial condition of daily based model is not mentioned. Please add some explanations. (3) Conclusion (2): "In the four catchments, by PAWN analysis of VMM, CS, IM, and KE are the most sensitive parameters and are not affected by the choice of objective functions, whereas WM is the most sensitive parameter" make me confusing. Please use clearer and more concise expressions. (4) Reference/citation style needs to be revised. For example, a space is missing between Lu and et on p5, line 26; parenthesis is not right on p7, line 4, (Pianosi and Wagener, 2015). (5) P3, line13:" Streamflow and rainfall data are from 1983 to 2009. Hourly streamflow data came from hydrological stations. Nine…", in this sentence, English tenses should be consistent. (6) P3, line 26: the runoff is conceptualized as being composed of surface runoff and groundwater flow (notoriously but erroneously called "below-ground off" in the paper). (7) Figure 1 is requested to be further processed. The symbols of rain gauge station and hydrological station are not very clear. (8) Figure 3: the y-axis label may be "absolute relative error of peak flow (%)" instead of "peak flow (%)". The title "Figure 3: Boxplot of peak flows . . ." should be also checked. (9) Although I am not a native English reviewer, I find some sentences difficult to understand. The authors are encouraged to further polish up the language.

---

## Referee Comment (RC2) · Anonymous Referee #2 · 17 Mar 2019

Dear authors,

I think you performed a lot work whose results deserve to be published. The topic addressed in your manuscript is interesting and I think that not only statistical hydrologists will benefit from its publication, either from the decision-makers point of view as well as from the perspective of semiarid catchments. However, I noticed several instances in your manuscript that force me not to accept it for publication in its current state.

First, I would like to emphasize that all my detailed comments are included in the

attached PDF file. Please, kindly see this attached file at the same scale, so you could find the places which my comments point to properly.

The paper is well structured, but a large number of style corrections is required.

Here are my general comments only:

The discussion of the results is not clear and, in my opinion, could be extended due to the huge effort made in the study. The authors are encouraged to extend the discussion of the results obtained by the explanation of possible causes of differences among hydrological models.

The initial condition is essential for this type of simulations. Thus, the authors should explain better this section.

Regarding the references shown in the manuscript, some are not listed in the reference list, and also there are some errors in the reference list. Please, check the standards of the journal and correct them.

I strongly recommend to undertake some check of the language (e.g. by some professional service offered directly by Springer on you websites). Without a doubt, there are some places in the text that are hardly understandable due to the language.

I believe my comments will help you improve your MS, which will lead to is final publication.

Please also note the supplement to this comment:
https://www.nat-hazards-earth-syst-sci-discuss.net/nhess-2018-402/nhess-2018-402-RC2-supplement.pdf

---

## Author Comment (AC1) · 10 Apr 2019

Answer to RC #1

General Comments: The paper presents a multicriterion assessment framework for flood events forecasting or warning in semi-arid regions. Four hydrologic models have been used in catchments of the middle Yellow River. The result shows that the VMM model has a better performance of flood modeling than the other three models. The authors believe that flood events in semi-arid and arid regions should have different

criteria than that of in humid areas to determine whether a flood forecasting and early warning is acceptable. The topic of this study is very interesting and the idea is more or less novel. The paper is well-structured. I recommend the publication of this paper with a minor revision. Some specific comments are listed as below.

Answer: Thank you for the positive comments on our manuscript.

Specific comments: (1) The proposed framework has three parts, C1, C2 and C3. C2 is the key part of this framework, and three flow zone (low flow zone, medium flow zone, high flow zone) are divided. I think this simple framework is more important in terms of flood early warning rather than just a performance assessment. This may be real value of the framework but it is not clearly expressed in the manuscript. More explanation and discussion should be added in the paper.

Answer: Thank you for this suggestion. We have added explanations and discussions to the manuscript, especially in Section 4, Results and discussion.

(2) The initial condition is very important for a hydrologic model. In this paper, it is reasonable that the daily based model is used to calculate the initial conditions of the event-based model, but the initial condition of daily based model is not mentioned. Please add some explanations.

Answer: Thank you for this good suggestion. We have rewritten the paragraph and added more detailed information about the initial condition. The revised paragraph is as follows: "The initial condition has important effects in modeling flood events. The VMM model was run continuously from 1983 to 2009 for each catchment. Two initial values are the initial tension water storage (W0) and the initial free water storage (S0) should be determined. Both of them represent the moisture content of the soil and were assumed to be zero due to the dry conditions at 00:00:00 on January 1, 1983. Rainfall data were available only at an hourly time step over the periods of flood events, and for other periods, they were available at a daily time step. Hence, the time step of simulation was daily between flood events and hourly within flood events."

(3) Conclusion (2): "In the four catchments, by PAWN analysis of VMM, CS, IM, and KE are the most sensitive parameters and are not affected by the choice of objective functions, whereas WM is the most sensitive parameter"make me confusing. Please use clearer and more concise expressions.

Answer: We have rewritten the sentence for clarity. "In the four catchments, the parameters confluence coefficient of surface flow (CS), impermeable area (IM), and residence time of Muskingum (KE) are the most sensitive based on an analysis by the global sensitivity method PAWN; in addition, the sensitivity ranking of the parameter WM related with the soil moisture capacity is the most affected by the objective functions."

(4) Reference/citation style needs to be revised. For example, a space is missing between Lu and et on p5, line 26; parenthesis is not right on p7, line 4, (Pianosi and Wagener,2015).

Answer: We have corrected this information.

(5) P3, line13:" Streamflow and rainfall data are from 1983 to 2009. Hourly streamflow data came from hydrological stations. Nine. . .", in this sentence, English tenses should be consistent.

Answer: Thank you for this good suggestion. We have changed this text.

(6) P3, line 26: the runoff is conceptualized as being composed of surface runoff and groundwater flow (notoriously but erroneously called "below-ground off" in the paper).

Answer: Thank you for this good suggestion. We have corrected this text.

(7) Figure 1 is requested to be further processed. The symbols of rain gauge station and hydrological station are not very clear.

Answer: We have improved the figure for clarity.

(8) Figure 3: the y-axis label may be "absolute relative error of peak flow (%)" instead of "peak flow (%)". The title "Figure 3: Boxplot of peak flows . . ." should be also

СЗ

checked.

Answer: We have corrected the label and checked the figure caption.

(9) Although I am not a native English reviewer, I find some sentences difficult to understand. The authors are encouraged to further polish up the language.

Answer: Thank you for this good suggestion. The manuscript has been polished by a professional service. All amendments are shown with tracked changes in the attached nhess-2018-402-supplement.pdf.

Please also note the supplement to this comment: https://www.nat-hazards-earth-syst-sci-discuss.net/nhess-2018-402/nhess-2018-402-AC1-supplement.pdf

---

## Author Comment (AC2) · 10 Apr 2019

Answer to RC #2

I think you performed a lot work whose results deserve to be published. The topic addressed in your manuscript is interesting and I think that not only statistical hydrologists will benefit from its publication, either from the decision-makers point of view as well as from the perspective of semiarid catchments. However, I noticed several instances in your manuscript that force me not to accept it for publication in its current state. First,

[Figure]

I would like to emphasize that all my detailed comments are included in the attached PDF file. Please, kindly see this attached file at the same scale, so you could find the places which my comments point to properly. The paper is well structured, but a large number of style corrections is required.

Answer: We thank the reviewer for the positive comments on our manuscript. Detailed style corrections are attached in the nhess-2018-402-supplement.pdf.

General Comments:

The discussion of the results is not clear and, in my opinion, could be extended due to the huge effort made in the study. The authors are encouraged to extend the discussion of the results obtained by the explanation of possible causes of differences among hydrological models.

Answer: Thank you for these constructive suggestions. We have added more explanation for clarity. Details are shown in the attached nhess-2018-402-supplement.pdf.

The initial condition is essential for this type of simulations. Thus, the authors should explain better this section.

Answer: Thank you for this good suggestion. We have written the paragraph and added some detailed information as follows:

"The initial condition has important effects in modeling flood events. The VMM model was run continuously from 1983 to 2009 for each catchment. Two initial values are the initial tension water storage (W0) and the initial free water storage (S0) should be determined. Both of them represent the moisture content of the soil and were assumed to be zero due to the dry conditions at 00:00:00 on January 1, 1983. Rainfall data were available only at an hourly time step over the periods of flood events, and for other periods, they were available at a daily time step. Hence, the time step of simulation was daily between flood events and hourly within flood events."

Regarding the references shown in the manuscript, some are not listed in the reference

list, and also there are some errors in the reference list. Please, check the standards of the journal and correct them.

Answer: We have checked all references and corrected the errors.

I strongly recommend to undertake some check of the language (e.g. by some professional service offered directly by Springer on you websites). Without a doubt, there are some places in the text that are hardly understandable due to the language.

Answer: Thank you for this good suggestion. The manuscript has been edited for language by a professional editing service. All amendments are shown with track changes in the attached nhess-2018-402-supplement.pdf.

I believe my comments will help you improve your MS, which will lead to is final publication.

Answer: Thank you. Your comments greatly helped us improve the manuscript, which we hope is now suitable for publication.

Please also note the supplement to this comment:
https://www.nat-hazards-earth-syst-sci-discuss.net/nhess-2018-402/nhess-2018-402-AC2-supplement.pdf

---

## Author Response (AR2)

**Comments to the Author:**

Dear authors,

On my previous decision (9 May), I have pointed the following major issues with your manuscript:

"The reviewers have generally positively evaluated your manuscript but also identified pertinent shortcomings, including (a) insufficient description of the model set-up, particularly regarding initialization (b) insufficient discussion and conclusions (c) reference list is missing many references and (d) language quality need to be strongly enhanced to make the content understandable. Other relevant points can be found in the reviewers comments."

You have re-submitted the manuscript on 10 May (!!!, one day later!!!) and you have not addressed properly at least three of the indicated four points, namely (a), (b), and (d). I am thus giving you a second chance to revise your manuscript in a way that properly addressed the raised concerns.

Please let me know if you would like to re-revise the manuscript. Please mark the additional changes to this version separately.

best regards
Joaquim Pinto
(handling editor)

**Response to the editor:**

Dear Dr. Joaquim Pinto,

Thank you for giving us the opportunity to revise the manuscript.

We are sorry that we did not properly address the major issues you mentioned last time. Actually, we began to revise our manuscript point by point according to the reviewers' comments at the end of March (after reviewers uploaded their comments). Therefore, we could quickly submit our revised manuscript after you gave us a positive comment. It is unfortunate that "at least three of the indicated four points" were not addressed properly in the previous manuscript.

This time, we rewrote Section 3.2 about model setup (including initialization) and have made it clearer. In addition, regarding the insufficient discussion and conclusions, we have provided more content in the manuscript, especially in Section 4. In Section 4, we expanded our discussion and rewrote Section 4.3; Table 5 has also been added to clarify our opinions. The manuscript has been edited for language by a professional editing service, and the edit certificate is attached below. We have also checked other points and answered the questions point by point again the reviewers

mentioned previously. All amendments are shown with tracked changes in the revised manuscript (below).

Addressing the identified issues truly helped us improve the manuscript, which we hope is now suitable for publication.

We are looking forward to your decision.

Best regards
Dayang Li, Zhongmin Liang, Yan Zhou, Binquan Li, Yupeng Fu

**AMERICAN JOURNAL EXPERTS**

**EDITORIAL CERTIFICATE**

This document certifies that the manuscript listed below was edited for proper English language, grammar, punctuation, spelling, and overall style by one or more of the highly qualified native English speaking editors at American Journal Experts.

**Manuscript title:**

Multicriteria assessment framework of flood events simulated with the vertically mixed runoff model in semiarid catchments in the middle Yellow River

**Authors:**

Dayang Li, Zhongmin Liang, Yan Zhou, Binquan Li, Yupeng Fu

**Date Issued:**

June 27, 2019

**Certificate Verification Key:**

5D57-6146-3608-39ED-402P

[Figure]

This certificate may be verified at www.aje.com/certificate. This document certifies that the manuscript listed above was edited for proper English language, grammar, punctuation, spelling, and overall style by one or more of the highly qualified native English speaking editors at American Journal Experts. Neither the research content nor the authors' intentions were altered in any way during the editing process. Documents receiving this certification should be English-ready for publication; however, the author has the ability to accept or reject our suggestions and changes. To verify the final AJE edited version, please visit our verification page. If you have any questions or concerns about this edited document, please contact American Journal Experts at support@aje.com.

American Journal Experts provides a range of editing, translation and manuscript services for researchers and publishers around the world. Our top-quality PhD editors are all native English speakers from America's top universities. Our editors come from nearly every research field and possess the highest qualifications to edit research manuscripts written by non-native English speakers. For more information about our company, services and partner discounts, please visit www.aje.com.

**Response to RC#1:**

**General Comments:**

The paper presents a multicriterion assessment framework for flood events forecasting or warning in semi-arid regions. Four hydrologic models have been used in catchments of the middle Yellow River. The result shows that the VMM model has a better performance of flood modeling than the other three models. The authors believe that flood events in semi-arid and arid regions should have different criteria than that of in humid areas to determine whether a flood forecasting and early warning is acceptable.

The topic of this study is very interesting and the idea is more or less novel. The paper is well-structured. I recommend the publication of this paper with a minor revision. Some specific comments are listed as below.

Answer: Thank you for the positive comments on our manuscript.

**Specific comments:**

(1) The proposed framework has three parts, C1, C2 and C3. C2 is the key part of this framework, and three flow zone (low flow zone, medium flow zone, high flow zone) are divided. I think this simple framework is more important in terms of flood early warning rather than just a performance assessment. This may be real value of the framework but it is not clearly expressed in the manuscript. More explanation and discussion should be added in the paper.

Answer: Thank you for this suggestion. We have almost rewritten Section 4 (Results and discussion) and added more explanations and discussions in the revised manuscript.

(2) The initial condition is very important for a hydrologic model. In this paper, it is reasonable that the daily based model is used to calculate the initial conditions of the event-based model, but the initial condition of daily based model is not mentioned. Please add some explanations.

Answer: Thank you for this good suggestion. The model was run continuously during the period of 1983-2009. The simulation was daily between flood events and hourly within flood events. Hence, there is no need to set initial values for each event. The revised paragraph is as follows:

"The VMM model was run continuously from 1983 to 2009 for each catchment. ... Two initial values, the initial tension water storage (W0) and the initial free water storage (S0), were used to describe the initial catchment moisture condition. The initial values are smaller for drier catchments, and the minimum values are zero. In this study, the initial values were assumed to be zero uniformly due to the dry conditions at 00:00:00 on January 1, 1983 for each catchment. It should be noted that continuous simulations for each catchment eliminate the need to set the initial values for each flood event in a catchment."

(3) Conclusion (2): "In the four catchments, by PAWN analysis of VMM, CS, IM, and KE are the most sensitive parameters and are not affected by the choice of objective functions, whereas WM is the most sensitive parameter" make me confusing. Please use clearer and more concise expressions.

Answer: We have rewritten the sentence for clarity.

"In the four catchments, the parameters confluence coefficient of surface flow (CS), impermeable

area (IM), and residence time of Muskingum (KE) in VMM model are the most sensitive based on

an analysis by the global sensitivity method PAWN; in addition, the sensitivity ranking of the parameter WM related with the soil moisture capacity is the most affected by the objective functions."

(4) Reference/citation style needs to be revised. For example, a space is missing between Lu and et on p5, line 26; parenthesis is not right on p7, line 4, (Pianosi and Wagener,2015).
Answer: We have corrected this information.

(5) P3, line13:" Streamflow and rainfall data are from 1983 to 2009. Hourly streamflow data came from hydrological stations. Nine. . .", in this sentence, English tenses should be consistent.
Answer: Thank you for this good suggestion. We have changed this text.

(6) P3, line 26: the runoff is conceptualized as being composed of surface runoff and groundwater flow (notoriously but erroneously called "below-ground off" in the paper).
Answer: Thank you for this good suggestion. We have corrected this text.

(7) Figure 1 is requested to be further processed. The symbols of rain gauge station and hydrological station are not very clear.
Answer: We have improved the figure for clarity.

(8) Figure 3: the y-axis label may be "absolute relative error of peak flow (%)" instead of "peak flow (%)". The title "Figure 3: Boxplot of peak flows . . ." should be also checked.
Answer: We have corrected the label and checked the figure caption.

(9) Although I am not a native English reviewer, I find some sentences difficult to understand. The authors are encouraged to further polish up the language.
Answer: Thank you for this good suggestion. The manuscript has been polished by a professional service. All amendments are shown with tracked changes in the attached pdf file.

**Response to RC#2:**

I think you performed a lot work whose results deserve to be published. The topic addressed in your manuscript is interesting and I think that not only statistical hydrologists will benefit from its publication, either from the decision-makers point of view as well as from the perspective of semiarid catchments. However, I noticed several instances in your manuscript that force me not to accept it for publication in its current state. First, I would like to emphasize that all my detailed comments are included in the attached PDF file. Please, kindly see this attached file at the same scale, so you could find the places which my comments point to properly.

The paper is well structured, but a large number of style corrections is required.

Answer: We thank the reviewer for the positive comments on our manuscript. Detailed style corrections are attached below.

**General Comments:**

The discussion of the results is not clear and, in my opinion, could be extended due to the huge effort made in the study. The authors are encouraged to extend the discussion of the results obtained by the explanation of possible causes of differences among hydrological models.

Answer: Thank you for these constructive suggestions. We have added more explanation for clarity.

The initial condition is essential for this type of simulations. Thus, the authors should explain better this section.

Answer: Thank you for this good suggestion. We have rewritten Section3.2 and added more detailed information as follows:

"The VMM model was run continuously from 1983 to 2009 for each catchment. … Two initial values, the initial tension water storage (W0) and the initial free water storage (S0), were used to describe the initial catchment moisture condition. The initial values are smaller for drier catchments, and the minimum values are zero. In this study, the initial values were assumed to be zero uniformly due to the dry conditions at 00:00:00 on January 1, 1983 for each catchment. It should be noted that continuous simulations for each catchment eliminate the need to set the initial values for each flood event in a catchment."

Regarding the references shown in the manuscript, some are not listed in the reference list, and also there are some errors in the reference list. Please, check the standards of the journal and correct them.

Answer: We have checked all references and corrected the errors.

I strongly recommend to undertake some check of the language (e.g. by some professional service offered directly by Springer on you websites). Without a doubt, there are some places in the text that are hardly understandable due to the language.

Answer: Thank you for this good suggestion. The manuscript has been edited for language by a professional editing service. All amendments are shown with track changes in the attached pdf file

I believe my comments will help you improve your MS, which will lead to is final publication.

Answer: Thank you. Your comments greatly helped us improve the manuscript, which we hope is now suitable for publication.

**Specific comments:**

**Page 1**

Line 10-12: It may be rewritten, although the sentence is understood, I think it is not the best way. Try to avoid double parentheses. Besides, the manuscript is focused on VMM, so this model should be first. E.g.: 'The ··· (VMM) is compared with three models, one physical based model, the MIKE SHE, and two conceptual models, the XAJ and the Shanbei' (this is just a suggestion, feel free to write your way).

Answer: We accept the reviewer's suggestion.

Change: The vertically mixed runoff model (VMM) is compared with three models, one physical-based model, MIKE SHE model (originated from the Système Hydrologique Européen program), and two conceptual models, the Xinanjiang model (XAJ) and Shanbei model (SBM).

Line 16: "its figure is only 41% in four semiarid catchments", could you explain better this sentence?

Answer: We apologize for our unclear expression, which has been revised.

Change: Our results show that the VMM model has a better flood estimation performance than the other models, and the FCRA framework can provide reasonable flood classification and reliability assessment information, which may help decision makers improve their diagnostic abilities in the early flood warning process.

Line 24: "and", but

Answer: This change has been made.

Change: but

**Page 2**

Line 9-10: Maybe it is better "to be used on a large-scale semiarid area".

Answer: Thank you for this good suggestion. We have changed this phrasing.

Change: The radar costs are too high to be used on a large-scale semiarid area.

Line 14: "severe", what is the meaning of severe here? It should be better explained.

Answer: "Severe rainstorms" means the rainstorms have produced the largest peak flows according to Michaud and Sorooshian (1994). We have added more explanation.

Change: Michaud and Sorooshian (1994) used 24 severe rainstorms that produced the largest peak flows during 1957-1977 to compare three hydrologic models, i.e., the lumped SCS model, simple distributed SCS model and distributed KINEROS model, in the Walnut Gulch catchment.

Line 20-21: "Therefore, it is urgent to search for useful information based on the limited accuracy of modeling results to serve as flood warnings and to improve decision making.". I do not understand this sentence. Please, rewrite the sentence.

Answer: We apologize for the unclear phrasing. We have rewritten the sentence.

Change: Therefore, determining how to use modeling results with limited accuracy to provide

guidance for flood early warning is important.

Line 24-25: "Four hydrologic models (the vertically mixed runoff model (VMM), MIKE SHE, Xinanjiang Model (XAJ) and Shanbei model (SBM)) ...". To much parenthesis; Try to make a continuous sentence: "Four hydrological models: the vertical ⋯, the MIKE SHE model, ⋯, are compared ...

Answer: Thank you for this suggestion. We have rewritten the sentence.

Change: Four hydrological models: the vertically mixed runoff model (VMM), the MIKE SHE model, Xinanjiang model (XAJ) and Shanbei model (SBM), are compared on the….

Line 27: "remainder". Please, use another word.

Answer: We have rewritten the sentence.

Change: The rest of the paper is organized as follows.

Line 27: "section below". Specify to which section corresponds exactly, so it is necessary to use capital letters: e.g. "The Methodology Section", or use instead of the name, just the number of the section "Section 3 describes⋯".

Answer: This is a good suggestion. We have changed this phrasing.

Change: Section 2 describes the study area and the data set used.

Line 28-31: "The VMM model...of the study.". Please rewrite the whole paragraph, it is difficult to follow it.

Answer: We have rewritten the paragraph to clarify our intent.

Change: Section 3 presents the VMM model methodology, model set, model calibration and validation, multicriteria assessment framework and parameter sensitivity analysis. Section 4 describes the results and discussion of model comparison, sensitivity analysis and analysis of the multicriteria assessment framework for the VMM model. The final section presents the conclusions of the study.

**Page 3**

Line 1: Is there any reference for all these data? I mean the temperature, rainfall and seasonality of the rainfall data.

Answer: The temperature data are obtained from previous papers, and we have added references. In addition, all rainfall data are collected from rain gauging stations, and we have added an explanation.

Change: This change has been made.

Line 3: "is", between.

Answer: This change has been made.

Change: between.

Line 4:" is", between.

Answer: This change has been made.

Change: between.

Line4: "65 – 80%", more less the 65 to 80%.

Answer: This change has been made.

Change: more less the 65 to 80%.

Line 5: "is", between.

Answer: This change has been made.

Change: between

Line 8: "soil erosion", how much?

Answer: Thank you for this good question. According to Li et al. (2019), the average sediment concentration reaches 126 kg m$^{-3}$ in these regions. We have added this information to the text.

Change: The lack of vegetation in these catchments leads to serious soil erosion, and the average sediment concentration reaches 126 kg m$^{-3}$ according to Li et al. (2019).

Line 9: "few attempts have been applied to model hourly flood flows". Please rewrite the sentence. It has poor writing.

Answer: We apologize for the poor phrasing. We have rewritten this sentence.

Change: Only a few studies have modeled hourly floods.

Line 11: "Hence, modeling floods and providing a useful method for decision makers in charge of flood defense are essential and urgent.". Please rewrite the whole sentence, I cannot follow it.

Answer: We apologize for the poor phrasing. We have rewritten this sentence.

Change: Hence, it is important for decision-makers to know how to evaluate the flood risk when a flood is approaching.

Line 12: "Streamflow and rainfall data are from 1983 to 2009." Please rewrite the sentence. It has poor writing.

Answer: We apologize for the poor phrasing. We have rewritten this sentence.

Change: The period used in the modeling is from 1983 to 2009. Streamflow and rainfall data are collected from streamflow gauging stations and rain gauging stations at an hourly time-step.

Line 12: "stations", which type of stations?

Answer: They are streamflow gauging stations. We have corrected this phrasing.

Change: Streamflow and rainfall data are collected from streamflow gauging stations and rain gauging stations at an hourly time-step, respectively.

Line 14-15: Thiessen polygon method was...

Answer: This change has been made.

Change: Thiessen polygon method was used to interpolate the rainfall data for each catchment.

Line 15: "...interpolate the rainfall data.". For each basin? I guess.

Answer: We agree with the reviewer. We have added this information.

Change: Thiessen polygon method was used to interpolate the rainfall data for each catchment.

Line 17: Here you should characterize the model in its whole, I mean, explain here also that it is lumped, continuous or event based, and so on.
Answer: Thank you for this suggestion. We have added this information.
Change: The VMM is a lumped continuous hydrologic model developed by Bao and Wang (1997).

Line 18: "...conceptual hydrologic model...", is continuous or event based?
Answer: It a continuous model; we have added this information.
Change: The VMM is a lumped continuous hydrologic model and has been used in many areas in China, especially in semiarid and subhumid catchments.

Line 21: "etc", better "and others".
Answer: Thank you for raising this good point. We have corrected this phrasing.
Change: This change has been made.

**Page 4**
Line 24: "in", at.
Answer: This change has been made.
Change: at.

Line 25-27: "Hence, tension water storage $W$... on daily values of $W$ and $S$". Please, explain it better, because I do not understand what type of variables you used in the case you just simulate a flood event.
Answer: We are apologized for our unclear phrasing. We have rewritten this paragraph.
Change: The VMM model was run continuously from 1983 to 2009 for each catchment. … Two initial values, the initial tension water storage (W0) and the initial free water storage (S0), were used to describe the initial catchment moisture condition. The initial values are smaller for drier catchments, and the minimum values are zero. In this study, the initial values were assumed to be zero uniformly due to the dry conditions at 00:00:00 on January 1, 1983 for each catchment. It should be noted that continuous simulations for each catchment eliminate the need to set the initial values for each flood event in a catchment.

**Page 5**
Line 1: "Because...". The sentence has poor writing. Maybe you it is better to write "Due to the fact that only ⋯ ". But I recommend rewriting the sentence.
Answer: Thank you for this good suggestion. We have rewritten the sentence.
Change: Due to the fact that only one streamflow gauging station is available for each catchment, the spatial variation in each catchment's parameters cannot be determined by calibration.

Line 1: "gauge", gauging.
Answer: This change has been made.
Change: gauging

Line 4: "Li et al., 2018", 2018a or 2018b?

Answer: We have checked the reference.

Change: Li et al., 2018

Line 8:" ...full fitness...". I guess you mean "fullfillnes", but I think you should better rewrite the whole sentence in order to get a continuity in the writing.

Answer: Thank you for raising this good point. We have rewritten this sentence.

Change: however, it may not be suitable for semiarid catchments because a good fit is not required between the simulated and observed streamflows.

Line 9: "(McIntyre and AI-Qurashi, 2009: SHARMA and MURTHY, 1998)". Remove parenthesis, write it properly. "McIntyre and Al-Qurashi (2008) and Sharma and Murphy (1998) used ⋯".

Answer: This change has been made.

Change: McIntyre and AI-Qurashi (2009) and Sharma and Murphy (1998) used…

Line 12: "$Q_{p'}^{i}$".  It is difficult to distinguish the apostrophe. Please, try another mark in order to make out the different variables

Answer: Thank you for this suggestion. We have changed this phrasing.

Change:  $Q_{pm}^{i}$.

Line 13: The same as previous.

Answer: Thank you for this suggestion. We have changed this phrasing.

Change:  $Q_{vm}^{i}$.

Line 15: Please, try another mark in order to make out the different variables.

Answer: This change has been made.

Change:$Q_{pm}$.

Line 16: The same as previous.

Answer: This change has been made.

Change: $Q_{vm}$

Line 20: "We...". It is up to you, but you should try to get coherence throughout the text. Consider using the impersonal form, as you have done in the rest of the manuscript.

Answer: Thank you for raising this good point. We have rewritten the sentence.

Change: The number of iterations was set to 2000 in the calibration process.

Line 21: "...step". Please, use another Word, as step can be misunderstood by iteration.

Answer: Thank you for raising this good point. We have changed this phrasing.

Change: The number of iterations was set to 2000 in the calibration process.

Line 26: "Wei-jian et al., 2016". It is not in the reference list.

Answer: We apologize. We have replaced it with another paper due to the reference being in Chinese.

Change: Cheng, C. T., Zhao, M. Y., Chau, K., and Wu, X. Y.: Using genetic algorithm and TOPSIS for Xinanjiang model calibration with a single procedure, J. Hydrol., 316, 129-140, 2006.

Line 26-27: "We test the performance... the middle Yellow River." This sentence is relevant? I mean, you should restructure the whole paragraph in order to get coherence, instead of specifying one by one each model.
Answer: Thank you for this good suggestion. We have made some changes.
Change: Please refer to the revised version of the manuscript.

Line 27: "Zhao (1983)". It is not in the reference list.
Answer: We have added this information.
Change: Zhao, R.: Watershed Hydrological Model: Xinanjiang Model and Shanbei Model, Water and Power Press, Beijing, China, 1983.

Line 28: "(Bao et al., 2017)". Could you provide more references?
Answer: We have added some references.
Change:
Li, Z. J., and Zhang, K.: Comparison of three GIS-based hydrological models, J. Hydrol. Eng., 13, 364-370, 2008.
Zhao, L., Xia, J., Xu, C. Y., Wang, Z., Sobkowiak, L., and Long, C.: Evapotranspiration estimation methods in hydrological models, J. Geogr. Sci., 23, 359-369, 10.1007/s11442-013-1015-9, 2013.

Line 28: "MIKE SHE...". Could you provide more characteristics of the model?
Answer: We have added more characteristics of the model.
Change: MIKE SHE originated from the Système Hydrologique Européen (SHE) program, and it is a deterministic, physically based distributed hydrologic model that can simulate surface water flow, unsaturated flow and saturated flow (Jayatilaka et al., 1998). MIKE SHE has been used to solve water resources and environment problems at different spatiotemporal scales (Li et al., 2018; Rujner et al.,2018; Samaras et al., 2016).

**Page 6**
line 3: There are many grammatical errors. Please, revise the whole point.
Answer: Thank you for pointing out these errors. We have corrected them.
Change: Please refer to the revised version of manuscript.

Line 3: "...of...". "for" instead of "of"?
Answer: This change has been made.
Change: for.

Line 4: "...characterizes...", characteristics.
Answer: This change has been made.
Change: characteristics.

Line 4: "and lack of enough rain gauges,", and also dispersion.

Answer: This change has been made.

Change: and also dispersion.

Line 5: "flood simulation", flood simulations.

Answer: This change has been made.

Change: flood simulations.

Line 6: "...to...", for.

Answer: This change has been made.

Change: for.

Line 7: "...flood feature...". I would remove "flood" as It is obvious.

Answer: This change has been made.

Change: This change has been made.

Line 9: "calculation...". I would remove this.

Answer: This change has been made.

Change: This word has been removed.

Line 11: "...and Bayesian method...". Which one? could you give some more information?

Answer: We have added this information.

Change: and Bayesian method with Markov chain Monte Carlo sampling.

Line 12: " but the way may...". I do not understand that phrase.

Answer: We have improved the sentence.

Change: although these methods may not lead to clear decisions.

Line 13: "...acquire...". Consider to change the Word.

Answer: This change has been made.

Change: obtain.

Line 13: "...utility...", useful.

Answer: This change has been made.

Change: useful.

Line 14: "...Yellow Rivers.". Please be careful. Remove "s".

Answer: This change has been made.

Change: ...Yellow River.

Line 16: "modeling...", modeled.

Answer: This change has been made.

Change: modeled

Line 16: "peak flows". Remove "s".

Answer: This change has been made.
Change: peak flow.

Line 21: "modeling...", modeled.
Answer: This change has been made.
Change: modeled.

Line 22: "(detailed ...)". I would not use the parenthesis here. Just use the semicolon.
Answer: Thank you for this good suggestion. We have improved the sentence.
Change: one component of the Bayesian forecasting system is detailed in Krzysztofowicz (1999) and Biondi et al. (2010).

Line 23: "(Krzysztofowicz, 1999; Biondi et al., 2010)". Remove parenthesis and write them in a correct way.
Answer: Thank you for noting these errors. We have corrected this phrasing.
Change: ···Krzysztofowicz (1999) and Biondi et al. (2010).

Line 25: "modelling peak flow...", the modelling peak.
Answer: This change has been made.
Change: the modelling peak.

**Page 7**
Line 4: "(Pianosi and Wagener, 2015)". Please write the parenthesis in the correct way. "pianosi and Wagener (2015) proposed ···".
Answer: Thank you for pointing out these errors. We have corrected this phrasing.
Change: Pianosi and Wagener (2015)

line 4: "PAWN", What is the meaning?. Explain the method.
Answer: PAWN is derived from the authors names according to Pianosi and Wagener (2015). We have added more explanation.
Change: Pianosi and Wagener (2015) proposed the novel GSA method PAWN (derived from the authors' names) based on the cumulative density function.

Line 5-6: "(Khorashadi Zadeh et al., 2017)". Please, write the reference in the correct way.
Answer: We have corrected this phrasing.
Change: Khorashadi et al. (2017)

Line 21: "5 events...". "five" instead of 5. Correct the rest of remaining ones throughout the paragraph.
Answer: We have corrected them.
Change: Five

Line 27: Results Section is good, but you have to rewrite it better, and expand it. Please, try to be more clear, and specify every result you have.

Answer: Thank you for these suggestions. We have rewritten Section 4.1.

Change: Please refer to the revised version of the manuscript.

**Page 8**

Line 1: "...VMM performs better...". In both calibration and validation? Please, expand your explanation.

Answer: We apologize for our unclear expression. We have rewritten the sentence.

Change: In terms of the median and average of the absolute relative errors for peak flows, except for the validation period in the Kuye River catchment shown in Figure 3 (h), Figures 3 (a)–(g) reveal that the VMM has lowest values for both calibration and validation.

Line 2: "...for both median and average peak flows." I understand what you are saying, but try to work out your explanation.

Answer: We have rewritten the sentence.

Change: In terms of the median and average of the absolute relative errors for peak flows, except for the validation period in the Kuye River catchment shown in Figure 3 (h), Figures 3 (a)–(g) reveal that the VMM has lowest values for both calibration and validation.

Line 11: "Overall...". Remove that. Rewrite the sentence.

Answer: We have rewritten the sentence.

Change: The analysis of Figure 3, Table 3 and Table 4 shows that the VMM has the best performance for flood modeling in the four studied catchments of the middle Yellow River.⋯

Line 11-12: "in the middle Yellow River". " of the" instead of "in".

Answer: This change has been made.

Change: of the

Line 12: "MIKE SHE...". There is a lack of connectors in some parts of the text. Here, you should use: "Besides", "In addition", etc.

Answer: This change has been made.

Change: In addition

Line 21: "Eq.(9) and Eq.(11)". Substitute the Equation by the variable, or write both of them. "Ep (Eq. 9) and Epv (Eq. 11)".

Answer: Thank you for this good suggestion. We have rewritten them.

Change: $E_p$(Eq. 9) and $E_{pv}$(Eq. 11).

Line 21-22: "The most sensitive ...objective functions.". Try to explain better what you want to mean. Rewrite the sentence.

Answer: We have rewritten the sentence.

Change: The higher the ranking is, the more sensitive the parameters. We can find that the parameters *CS*, *IM* and *KE* have the highest rankings whether the objective function of the VMM model is $E_p$ or $E_{pv}$.

Line 29: "...must meet...". There are two "must" in the sentence. Please, use synonym or rewrite the sentence.
Answer: We have rewritten the sentence.
Change: The framework requires that an accepted flood event should meet one of the requirements of C1 and C2; in addition, C3 needs to be satisfied simultaneously.

Line 30: "Flood events conforming to conditions...". Is that right?
Answer: We apologize for the unclear expression. We have rewritten the sentence.
Change: The observed peak flows and the modeled peak flows under the conditions C1, C2 or C3 are shown in Figure 5.

**Page 9**
Line 15: "...VMM...". In the conclusions part, you should rewrite the abbreviations. Please, check the rest of references and abbreviations.
Answer: We have checked them and added an explanation for clarification.
Change: Please refer to the revised version of the manuscript.

Line 19-20: "In the four catchments, by...sensitive parameter.". The same as previous.
Answer: We have rewritten the sentence.
Change: In the four catchments, the parameters confluence coefficient of surface flow (*CS*), impermeable area (*IM*), and residence time of Muskingum (*KE*) are the most sensitive based on an analysis by the global sensitivity method PAWN; in addition, the sensitivity ranking of the parameter *WM* related with the soil moisture capacity is the most affected by the objective functions.

Line 23: "The condition C2... ". Explain that. You should bear in mind Conclusions could be read by anyone, so they should not contain references to the rest of the article unless it is completely necessary.
Answer: We have added more explanation.
Change: The condition C2, which divides peak flows into three flow zones, will be affected…

Line 24: "... enrich...". Please, use another word.
Answer: This change has been made.
Change: The framework… provide guidance for decision making.

**Page 10**
Line 5: "Reference:". There are several errors in the reference list. Please, read the journal's rules, and correct every mistake. I just write down some of them. Besides, order the references according to the standards of the journal.
Answer: We have checked the references and corrected them.

Change: Please refer to the revised version of the manuscript.

Line 6: "Andersen....". This is a PhD Thesis. Please, read journal's rules.
Answer: This change has been made.
Change: Andersen, F. H.: Hydrological modeling in a semi-arid area using remote sensing data, Ph.D. thesis, University of Copenhagen, Copenhagen, Denmark, 2008.

Line 8-9: "Burnash...". I found the paper, but there is a lack of information here. Please complete the reference.
Answer: This change has been made.
Change: Burnash, R. J., Ferral, R. L., and McGuire, R. A.: A generalized streamflow simulation system, conceptual modeling for digital computers, Report by the Joliet Federal State River Forecasts Center, Sacramento, CA, 204 pp., 1973.

Line 11-12: "Bao, W. and Wang, C....". Unless, it is completely necessary, please, try to avoid references that are hard to find out. I could not find those references in Chinese.
Answer: Thank you. We believe that this suggestion is very constructive. We have replaced this reference with another paper. However, some references in Chinese are completely necessary, so we have kept them.
Change: Wang, G., and Ren, L.: A Contrastive Study of Simulation Results between GWSC-VMR and Hybrid Runoff Model in Dianzi Basin, in: International Conference on Environmental Science and Information Application Technology, Wuhan, China, 4–5 July, 583-588, 2009.

Line 22: "Beven, K.:...". It is needed more info about these references.
Answer: This change has been made.
Change: Beven, K. J.: Environmental modelling: An uncertain future?, CRC press, London, UK, 328 pp., 2007

Line 23: "Beven, K.:...". It is needed more info about these references.
Answer: This change has been made.
Change: Beven, K. J.: Rainfall-runoff modelling: the primer, John Wiley & Sons, UK, 488 pp., 2011

**Page 11**
line 4: "3-23, 2007." DOI?
Answer: This change has been made.
Change: Collier, C. G.: Flash flood forecasting: What are the limits of predictability?, Q. J. Roy. Meteor. Soc., 133(622), 3–23, https://doi.org/10.1002/qj.29, 2007

Line 31-32: "Li, D: ...". Rewrite according to the journal's rules.
Answer: This change has been made.
Change: Li, D.: Hydrologic model: the vertically mixed runoff model (vmm), HydroShare, https://doi.org/10.4211/hs.c523228 7d5c04bfb8cac5ce4e391ea 0f, 2018

**Page 12**

Line 2: "...Qiushui River, Yellow River, 06, 24-28, 2018a.". This is a journal?? Please, again try to avoid these references, but also try to make easy to find them in case you want to add them to the list.

Answer: We have deleted it.

Change: The reference has been deleted.

Line 29: "1-32". Pages range is wrong.

Answer: Thank you for noting these errors. We have corrected this phrasing.

Change: Sharma, K. D., and Murthy, J. S. R.: A practical approach to rainfall-runoff modelling in arid zone drainage basins, Hydrolog. Sci. J., 43(3), 331 – 348, 1998

Line 32: "United Nations Environment...". More information about the Book.

Answer: This change has been made.

Change: United Nations Environment Programme (UNEP): World Atlas of Desertification, Edward Arnold, London, 69 pp., 1992

**Page 13**

Line 3: "...Yellow River.". The same as previous.

Answer: This change has been made.

Change: Please refer to the revised manuscript.

Line 8: "Mathematical modelling and computational experiments.", abbreviation?

Answer: This change has been made.

Change: Sobol, I. M.: Sensitivity estimates for nonlinear mathematical models, Math. Model. Comput. Exp., 1, 407 – 414, 1993.

Line 13: "44(5), 2008.", pages range?

Answer: We have added this information.

Change: Yatheendradas, S., Wagener, T., Gupta, H., Unkrich, C., Goodrich, D., Schaffner, M. and Stewart, A.: Understanding uncertainty in distributed flash flood forecasting for semiarid regions, Water Resour. Res., 44(5), 61 – 74, 2008.

Line 15: "J.GEOPHYS. RES-ATMOS.". Why in capital letters?

Answer: Thank you for noting these errors. We have corrected this phrasing.

Change: Young, C.B., Nelson, B.R., Bradley, A.A., Smith, J.A., Peters-Lidard, C.D., Kruger, A. and Baeck, M.L.: An evaluation of NEXRAD precipitation estimates in complex terrain, J. Geophys. Res.-Atmos., 104, 19691 – 19703, 1999.

**Page 18**

Figure 1: What is the meaning of the big R? Please, include other references in the map of the Yellow River Basin, maybe the border of the see or the cities included inside the basin. Also, the rain gaunges stations are not clear. Please, change the mark. Finally, the drainage basin is too gross. Please, try to add some more detail.

Answer: Thank you for these suggestions. The big R may be a display error. It should be the North Arrow. In addition, we have checked all figures in case other errors occurred. We have improved Figure 1 for clarity.
Change:

[Figure]

Figure 1: Location of the Qingjian River catchment, Qiushui River catchment, Tuwei River catchment and Kuye River catchment.

**Page 20**

Figure 3: Please, add letters to identify each graph, in order to be able to talk about them in the manuscript. Also, you should add some space among graphs. Regarding the axes, specify clearly what is the variable they are measuring, I guess they talk about errors.

Answer: Thank you for these suggestions. We have improved Figure 3. In addition, the Y axis label should be "Absolute relative errors of peak flows (%)". We have corrected this phrasing.

Change:

[Figure]

Figure 3: Boxplots of the absolute relative errors of the peak flows in the four catchments; Q1 and Q3 resprent the first quantile and third quantile, respectively; interquartile range (IQR)= $Q3 - Q1$; and an outlier is defined as an extreme value that exceeds the IQR.

**Page 21**

Figure 4: Please, rewrite the whole figure caption. It is very difficult to understand what you try to say.

Answer: We apologize for the lack of clarity. We have rewritten the figure caption.

Change: Figure 4: Sensitivity rankings of the VMM parameters based on the global sensitivity analysis method PAWN for different objective functions: (a) $E_{pv}$ as the objective function, and (b) $E_p$ as the objective function. The value $P$ is used to assess the sensitivity degree of the parameter with the PAWN method, and a larger value corresponds to greater sensitivity. The numbers on the ordinate represent the sensitivity rankings.

**Page 22**

figure 5: Please, add letters at each graph, in order to be able to talk about them in the text. In addition, clarify the legend, it is difficult to identify each variable. Regarding the caption, please give more info, more detail about the graph. Explain more about we are watching.

Answer: Thank you for this good suggestion. We have added letters at each graph and additional explanations in the figure caption. In addition, additional explanations have been added to Section 4.3.

Change:

[revised manuscript text omitted]

---

## Author Response (AR3)

**Comments to the Author:**

Dear author,

Your manuscript has been re-revised by both original referees. Based on their and my own evaluation, the manuscript is returned to technical and mandatory corrections.

Provided that this is carefully done, the manuscript will then be accepted for publication.

best regards

Joaquim Pinto (handling editor)

**Response to the editor:**

Dear Dr. Joaquim Pinto,

We would like to thank the editor and the reviewers for helping us improve the manuscript for publication.

We have revised our manuscript point by point carefully according to the reviewers' suggestions. Our responses to the reviewers and a marked-up manuscript version are attached below.

Thank you again.

Best regards

Dayang Li, Zhongmin Liang, Yan Zhou, Binquan Li, Yupeng Fu

**RC#1**

My suggestion is accepted subject to technical corrections.

Answer: Thank you very much. We appreciate your time for improving our manuscript.

**RC#2:**

Some specific comments:

- I do not understand why you write quotation marks on classification-reliability throughout the text.

Answer: We apologize for this problem. We have deleted the quotation marks throughout the text.

- I would try to avoid as much parenthesis as possible, such as in page 2 line 17 to line 19, or when you are citing figures. Please, rewrite all Figure citations in the text without parenthesis (E.g. in page 8 line 20: "in Fig. 3a and 3g" or in page 8 line 22: "Fig. 3a, b, c, d, g and h.)

Answer: Thank you for this suggestion. We have checked the whole text to avoid your mentioned problems.

- In page 5 line 15, change the word realized.

Answer: we have changed the word "realized" as "achieved".

- In page 6 line 10, remove the second semicolon.

Answer: This change has been made.

- In page 6 line 17, Change part of the sentence "Due to strong spatial variability..."

Answer: This change has been made:

" Flood simulations and forecasting in semiarid catchments are very difficult due to strong spatially the variability of rainfall, complex landscape characteristics and others."

- In page 6 line 29, add some indicators when you are citing the two parts, e.g. "The FCRA framework consists on two parts: i) flood classification and ii) flood reliability assessment."

Answer: Thank you for this good suggestion. This change has been made.

- In page 8 line 24, add "the" before "other catchments".

Answer: This change has been made.

- In page 8 line 24, you should explain to which range you are referring. Please, explain it not only in the figure caption but also in the text.

Answer: This change has been made: "In terms of interquartile ranges (IQR) of the absolute relative errors for peak flows...".

- In page 9 line 13, please rewrite the sentence "this simulation ... conceptual models".

Answer: Thanks for your suggestion. We have rewritten the sentence:" it may be the reason why it performs better than the other two conceptual models".

- In page 10 line 25, add "a" before "better performance".

Answer: This change has been made.

- In page 11 line 26, separate the references of Beven 2011 and Beven 2001.

Answer: This change has been made.

- In page 12 line 7 and line 22, change the hyphen for a dash.

Answer: This change has been made.

- In page 13 line 9, remove the space in the end of the DOI number

Answer: This change has been made.

- In page 14 line 20, add the year the pages and the volume of the reference.

Answer: This change has been made.

- In Table 2, change the caption, as I think the values of the parameters are just recommendations from other authors and not the calibrated parameters of this paper.

Answer: We agree with the reviewer. We have changed the caption as "Parameter values of the VMM".

- In Figure 1, I would add some other information in the caption, such as the raingauging net.

Answer: We have added the raingauging net in the caption of Figure 1.

- In Figure 3, please rewrite the last sentence of the Figure caption. Separate the sentence by a dot and not only by a comma, and also reference the outlier mark in the Figure.

Answer: I have rewritten the sentence: "An outlier is defined as an extreme value that exceeds the range of Q1-1.5 IQR and Q3+1.5IQR".

In Figure 5, in the first sentence of the caption, when you are explaining the simulated peak flows, I would write just "circles" and then explain why some are solid and the others are empty, because after that you only explain the meaning of the solid ones.

Answer: Thank you for this suggestion. We have rewritten the sentence.

Besides these comments, I have some other general comments.

- As noted in the previous review, there are (now more than before) some parts of the text in impersonal form, while other paragraphs are in personal form (page 7 lines 13 and 17, page 10 line 8). I would recommend maintaining the coherence in the text.

Answer: Thank you for this good suggestion. We have checked the whole text and made changes.

- Besides, although you have corrected many errors in the references, I recommended to add as much information as possible in order to track the papers you are referring. So, it is very important to add the DOI number whenever is possible. Please, add more doi numbers to the references, specially to those who are easy to find.

Answer: We agree with the reviewer. We have added DOI numbers to the majority of the references.

[revised manuscript text omitted]

(9)

$$E_{\nu} = \frac{1}{n} \sum_{i=1}^{n} \frac{|Q_{\nu}^{i} - Q_{\nu m}^{i}|}{Q_{\nu m}^{i}}$$
(10)

where  $E_p$  and  $E_v$  are the average performances (in terms of absolute relative error) for peak flows and flow volumes in each catchment, respectively; *n* is the number of events; the index *i* denotes each event;  $Q_p$  and  $Q_{pm}$  are the simulated and measured values of peak flow per event, respectively; and  $Q_v$  and  $Q_{vm}$  are the simulated and measured values of flow volume per event,

|   | Deleted: realized |
|---|-------------------|
| - | Deleted: )(       |
|   |                   |

respectively.

Constraining the model output with peak flows and flow volumes can be expressed as follows:

$$E_{pv} = \frac{E_p + E_v}{2} \tag{11}$$

where  $E_{pv}$  is the objective value. The model outputs become better as the value of  $E_{pv}$  approaches 0. The number of iterations was set to 2000 in the calibration process.

**3.4 Model comparison**

To achieve a better performance in rainstorm flood simulations, three hydrologic models, including two conceptual models, XAJ and SBM, and one distributed model, MIKE SHE, were used for comparison with the VMM model. The XAJ model was developed by Zhao, (1992) and has a single saturation excess runoff generation mechanism. The XAJ model has been successfully applied in humid and subhumid catchments (Cheng et al., 2006; Lü et al., 2013). The SBM model was developed by Zhao (1983) and has a single infiltration excess runoff generation mechanism. The SBM model was developed by Zhao (1983) and has a single infiltration excess runoff generation mechanism. The SBM model is generally used in semiarid or arid catchments in China (Bao et al., 2017; Li and Zhang, 2008; Zhao et al., 2013). In addition, the MIKE SHE model is a deterministic, physically-based distributed hydrologic model that can simulate surface water flow, unsaturated flow and saturated flow (Jayatilaka et al., 1998). The MIKE SHE model has been used to solve water resources and environment problems at different spatiotemporal scales (Li et al., 2018; Rujner et al., 2018; Samaras et al., 2016).

15

20

25

30

10

5

**3.5 Multicriteria assessment framework: flood classification-reliability, assessment for flood events**

Flood simulations and forecasting in semiarid catchments are very difficult due to strong spatially the variability of rainfall, complex landscape characteristics and others. Although some hydrologists improve flood simulations and forecasting by improving hydrologic models, the improvements are always limited or are suitable for only specific regions (Collier, 2007). The flood peak is the most significant feature in semiarid regions. Determining the extent to which the calculation of flood peaks can be accepted is crucial. Generally, the absolute relative error is used to measure the calculation of flood peak accuracy; for example, 20%, 30% or similar values are acceptable (Li et al., 2014; McIntyre and Al-Qurashi, 2009). To provide more information for flood defense management, the generalized likelihood uncertainty estimation (GLUE) and the Bayesian framework with Markov Chain Monte Carlo sampling are used to provide probabilistic forecasting, such as the 95% uncertainty interval (Christiaens and Feyen, 2002; Li et al., 2017), although these methods may not lead to clear decisions (Beven, 2007).

In this study, to obtain a better diagnostic and discriminatory method for the decision maker, we propose a multicriteria assessment framework called the flood classification—reliability assessment (FCRA) in the catchments of the middle reaches of the Yellow River. The FCRA framework consists two parts: i) flood classification and ii) flood reliability assessment. The first part represents floods are classified with percentiles and the absolute relative error; the other represents the reliability of

| 1 | Deleted: ( |
|---|------------|
| 1 | Deleted: , |
| 1 | Deleted: ; |

|   | Deleted: " |  |
|---|------------|--|
|   | Deleted: " |  |
| - |            |  |

| _ | Deleted: " |
|---|------------|
|   | Deleted: " |
| _ | Deleted: ; |

flood modeling is evaluated with the Bayesian method. Peak flows, as the most prominent features of flood events, are assessed with the FCRA framework. Detailed descriptions can be found as follows:

(C1) The absolute relative error of peak flow should be less than 20%.

(C2) The modeled and observed peak flows should be in the same flow zone: the observed peak flow  $Q_p$  for all flood events in a catchment are divided into three zones (low flow zone, medium flow zone, high flow zone), with 25th percentiles  $Q_{p25}$  and 75th percentiles  $Q_{p75}$  as the boundary points; if  $Q_p \leq Q_{p25}$ , then the peak flow  $Q_p$  belongs to the low flow zone; if  $Q_p \geq Q_{p75}$ , then the peak flow  $Q_p$  belongs to the high flow zone; the remaining flow peaks belong to the medium flow zone. Both the 25th percentile and 75th percentile are commonly used to distinguish zones.

(C3) The observed peak flows should fall within one standard deviation (σ) of the mean (approximately 68.3% uncertainty
 interval) peak flow estimated by the Hydrologic Uncertainty Processor (HUP), one component of the Bayesian forecasting system detailed in Krzysztofowicz (1999) and Biondi et al. (2010).

Conditions C1 and C2 are flood classification criteria. If the observed and modeled peak flows meet one of the two conditions, it isbelieved that they are the same types of floods. The key of the FCRA framework is condition C2, and condition C1 is used to avoid errors caused by flow zone boundaries. For example, when  $Q_{p75} = 200 \text{ m}^3$ /s, the modeled peak flow equals 198 m3/s and the observed peak flow equals 201 m3/
[revised manuscript text omitted]
 <a href="https://doi.org/10.4211/hs.c5232287d5c04bfb8cac5ce4e391ea0f">https://doi.org/10.4211/hs.c5232287d5c04bfb8cac5ce4e391ea0f</a>. Field Code Changed 5 Author contributions DL wrote the text and developed the MATLAB code for the VMM model. DL, ZL, YZ and BL conceived the study. All co-Formatted: Line spacing: single authors jointly worked on improving the text and responded to the editor's and the reviewers' suggestions, Formatted: Pattern: Clear **Competing interests** The authors declare that they have no conflict of interest, Formatted: Font: 10 pt Formatted: Line spacing: single Formatted: Font: Not Italic 10 Acknowledgments This study was supported by the National Key Research and Development Program of China (grant no. 2016YFC0402706), the National Natural Science Foundation of China (grant nos. 41730750, 41877147), and Special Scientific Research Fund of Public Welfare Industry of Ministry of Water Resources, China (201501004), sponsored by Qing Lan Project. We would like to thank Francesca Pianosi (University of Bristol) for providing the program code of PAWN at https://www.safetoolbox.info Formatted: Default Paragraph Font, Font: (Asian) 等线, 10.5 pt 15 /pawn-method/. We also thank the editor and the anonymous reviewers, whose comments have largely improved this work. References: Andersen, F. H.: Hydrological modeling in a semi-arid area using remote sensing data, Ph.D. thesis, University of Copenhagen, Copenhagen, Denmark, 2008. Bao, H., Wang L., Zhang, K. and Li, Z.: Application of a developed distributed hydrological model based on the mixed runoff 20 generation model and 2D kinematic wave flow routing model for better flood forecasting, Atmos. Sci. Lett., 18(7), 284-293, https://doi.org/10.1002/asl.754, 2017. Formatted: Default Paragraph Font, Font color: Black Bao, W.: Improvement and application of the Green-Ampt infiltration curve, Yellow River, 9, 1-3, 1993. (In Chinese) Bao, W. and Zhao, L.: Application of Linearized Calibration Method for Vertically Mixed Runoff Model Parameters, J. Hydrol. Eng., 33(4), 85-91, https://doi.org/10.1061/(ASCE)HE.1943-5584.0000984, 2014. 25 Beven, K. J.: Surface water hydrology-runoff generation and basin structure, Rev. Geophys., 21(3), 721-730, https://doi.org Formatted: Default Paragraph Font, Font color: Black /10.1029/RG021i003p00721, 1983. Beven, K. J.: Towards an alternative blueprint for a physically based digitally simulated hydrologic response modelling system,

[revised manuscript text omitted]